# Vascular dimorphism ensured by regulated proteoglycan dynamics favors rapid umbilical artery closure at birth

Sumeda Nandadasa[1], Jason M Szafron[2], Vai Pathak[3], Sae-Il Murtada[2], Caroline M Kraft[1], Anna O'Donnell[1], Christian Norvik[4], Clare Hughes[5], Bruce Caterson[5], Miriam S Domowicz[6], Nancy B Schwartz[6], Karin Tran-Lundmark[4], Martina Veigl[3,7], David Sedwick[7], Elliot H Philipson[1,8], Jay D Humphrey[2]*, Suneel S Apte[1]*

[1]Department of Biomedical Engineering, Cleveland Clinic Lerner Research Institute, Cleveland, United States; [2]Department of Biomedical Engineering, Yale University, New Haven, United States; [3]Case Comprehensive Cancer Center, Case Western Reserve University, Cleveland, United States; [4]Department of Experimental Medical Science and Wallenberg Center for Molecular Medicine, Lund University, Lund, Sweden; [5]The Sir Martin Evans Building, School of Biosciences, Cardiff University, Cardiff, United Kingdom; [6]Department of Pediatrics, University of Chicago, Chicago, United States; [7]Department of Medicine, Case Western Reserve University, Cleveland, United States; [8]The Women's Health Institute, Department of Obstetrics and Gynecology, Cleveland Clinic, Cleveland, United States

*For correspondence:
jay.humphrey@yale.edu (JDH);
aptes@ccf.org (SSA)

Competing interests: The authors declare that no competing interests exist.

**Abstract** The umbilical artery lumen closes rapidly at birth, preventing neonatal blood loss, whereas the umbilical vein remains patent longer. Here, analysis of umbilical cords from humans and other mammals identified differential arterial-venous proteoglycan dynamics as a determinant of these contrasting vascular responses. The umbilical artery, but not the vein, has an inner layer enriched in the hydrated proteoglycan aggrecan, external to which lie contraction-primed smooth muscle cells (SMC). At birth, SMC contraction drives inner layer buckling and centripetal displacement to occlude the arterial lumen, a mechanism revealed by biomechanical observations and confirmed by computational analyses. This vascular dimorphism arises from spatially regulated proteoglycan expression and breakdown. Mice lacking aggrecan or the metalloprotease ADAMTS1, which degrades proteoglycans, demonstrate their opposing roles in umbilical vascular dimorphism, including effects on SMC differentiation. Umbilical vessel dimorphism is conserved in mammals, suggesting that differential proteoglycan dynamics and inner layer buckling were positively selected during evolution.

## Introduction

The umbilical cord, typically containing two arteries and one vein in humans, is a crucial fetal structure in placental mammals. Umbilical arteries carry fetal blood to the placental vascular bed, whereas the umbilical vein returns oxygenated blood to the fetus. Neonatal respiration at birth renders the maternal oxygen supply redundant. Umbilical arteries commence closure rapidly after delivery of the newborn whereas the veins remain open longer. The cord is routinely clamped following delivery and divided between the clamps in modern obstetric practice. Timing of cord clamping after birth, whether early or late, is extensively debated (*Niermeyer, 2015*; *Tarnow-Mordi et al., 2017*). A recent recommendation suggested clamping no earlier than 30–60 s after birth to facilitate the

placental transfusion (*TACoOaG, 2017*). Although the necessity of clamping is rarely questioned, it appears to be a modern practice (*Downey and Bewley, 2012*). Cord clamping is rarely practiced in domesticated animals and certainly not in wild animals, yet all current mammalian species have survived evolutionarily. We hypothesized that intrinsic design characteristics of mammalian umbilical arteries prevent blood loss at birth without clamping.

Prior histological work revealed that umbilical arteries have a bilaminar structure (*Meyer et al., 1978*) but lack elastic lamellae, which endow large arteries with resilience during cyclic loading (*Wagenseil and Mecham, 2009*). However, the molecular mechanism underlying the bilayered structure and its relationship to arterial occlusion remains obscure. Here, we used a multi-disciplinary approach integrating a variety of morphologic approaches with mechanical testing, computational analysis and mouse mutants to demonstrate the molecular and biomechanical basis for rapid umbilical artery closure. The findings emphasize the dual importance of extracellular matrix proteoglycans in regulation of cell differentiation and conferment of desirable tissue mechanical characteristics.

## Results

### The umbilical artery has a bilaminar wall

Three-dimensional imaging of term human umbilical cords, using synchrotron-based phase contrast micro-CT with effective pixel size $1.63 \times 1.63$ µm$^2$ (*Norvik et al., 2020*) and histology, identified a much thicker tunica media (TM) in the umbilical artery than in the vein, with a visibly different structure (*Figure 1a–c*, *Figure 1—figure supplement 1a–c*, *Figure 1—videos 1*, *2*). Most umbilical arteries were occluded at birth independent of delivery method or cord region analyzed, whereas umbilical veins remained patent (*Figure 1b*, *Figure 1—figure supplement 1b*). Smooth muscle cell (SMC) markers showed similar staining intensities within inner and outer TM of the umbilical arteries and TM of the vein with alternating layers of longitudinal and circumferentially oriented SMCs (*Figure 1—figure supplement 1c,d*). The veins showed fewer layers of SMCs compared to the arteries (*Figure 1—figure supplement 1c,d*).

Alcian blue, which binds sulfated glycosaminoglycans (GAGs), intensely stained the inner layer of the bilayered arterial TM but only the innermost three to four cell layers of the venous TM (*Figure 1c*). SMCs in this GAG-rich region of the arteries were radially oriented and round, with nuclear-localized Sox9, a chondrogenic factor (*Figure 1—figure supplement 1e*; *Ng et al., 1997*). The distribution of chondroitin sulfate (CS) coincided with Alcian blue staining (*Figure 1c–d*, *Figure 1—figure supplement 1e*), whereas heparan sulfate was more abundant in the outer arterial TM (*Figure 1d*), suggesting that the inner TM was enriched in CS-proteoglycans (CSPGs). RNA microarray data from matched human umbilical arteries and veins showed, among many differentially expressed genes (*Figure 1e*, *Figure 1—figure supplement 2*, Sup. array data-1), arterial prevalence of mRNAs for *ACAN* and *VCAN* encoding CSPGs bearing the most CS-chains, aggrecan and versican, respectively (*Figure 1e*). Microarray analysis of the inner versus outer arterial TM identified stronger *ACAN* and *VCAN* expression in the inner TM, among other differences (*Figure 1e*, *Figure 1—figure supplement 3*, Sup. array data-2). RNA in situ hybridization (RNA-ISH) localized strong *ACAN* and *VCAN* expression in inner arterial TM SMC, and immunostaining showed versican and aggrecan core proteins in a similar distribution as alcian blue and anti-CS staining (*Figure 1c,d, f*). Versican is a well-characterized vascular component (*Wight and Merrilees, 2004*), and aggrecan, which is known as a cartilage and neural proteoglycan (*Lauing et al., 2014*; *Schwartz and Domowicz, 2014*), is emerging as a significant CSPG in vascular disease (reviewed in *Koch et al., 2020*).

### ADAMTS proteoglycanases are differentially expressed in the umbilical artery and vein

Aggrecan and versican are proteolytically cleaved by ADAMTS1, 4, 5, and 9 (*Dancevic et al., 2016*). *ADAMTS1* and *ADAMTS4* mRNAs had higher levels in the venous wall in microarrays (*Figure 1e*, Sup. Array data-1), and RNA-ISH demonstrated stronger expression of *ADAMTS1*, *ADAMTS4*, *ADAMTS5,* and *ADAMTS9* in the veins (*Figure 2a*). *ADAMTS1* was the most strongly expressed, localizing to venous endothelium and TM, with stronger umbilical artery expression seen in the outer than inner TM (*Figure 2a*). *ADAMTS9* was similarly expressed as *ADAMTS1*, whereas *ADAMTS4* and *ADAMTS5* mRNAs were restricted to umbilical vein endothelium and some venous SMC (*Figure 2a*).

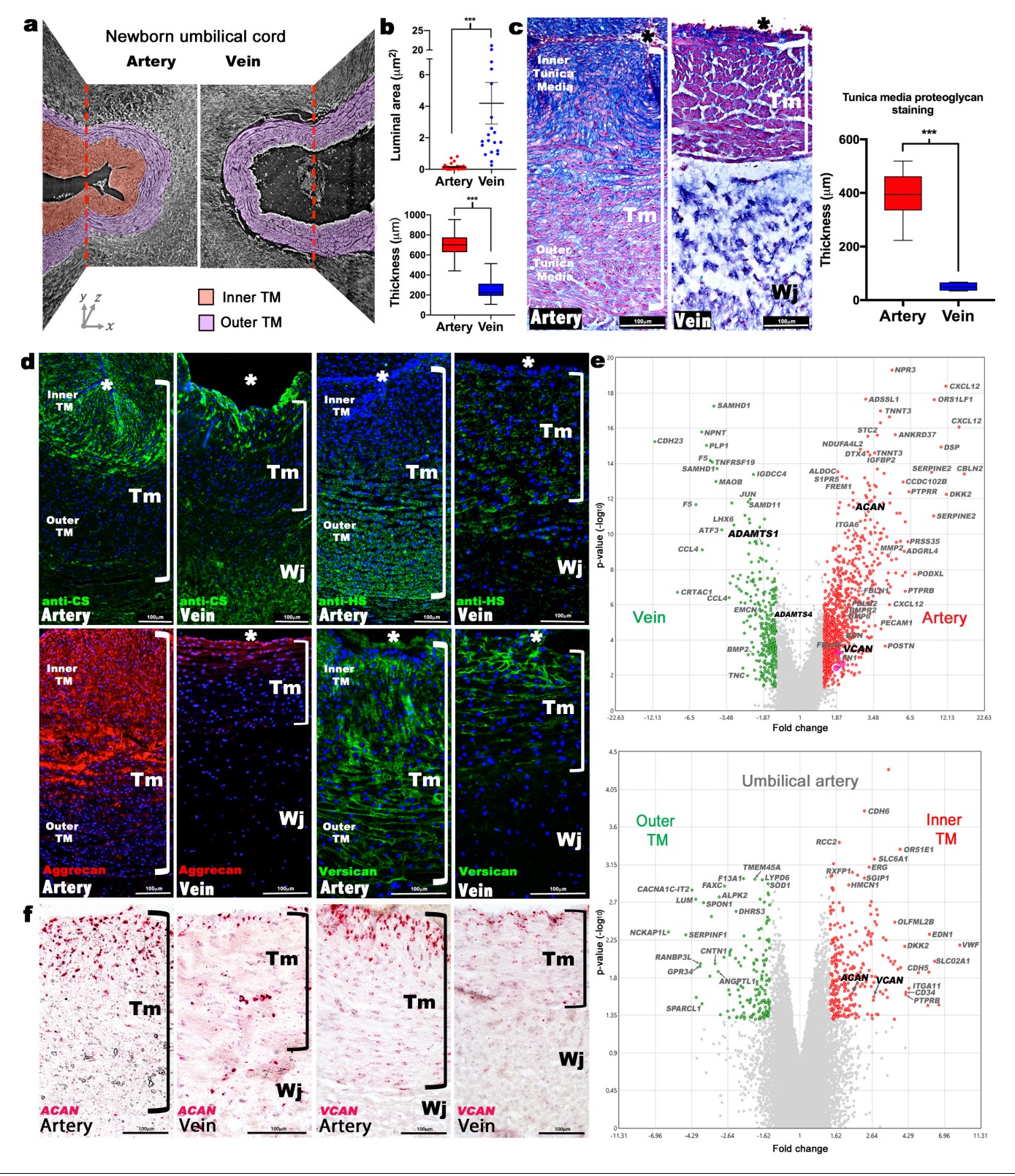

**Figure 1.** Dimorphism of the human umbilical artery and vein. (a) Synchrotron imaging of umbilical vessels at birth illustrates a bilayered arterial wall comprising an inner buckled tunica media (TM) (red) and outer TM (purple) but no distinct inner layer or buckling in the vein. X-Y and Y-Z image planes are indicated by red dashed lines (n = 3 umbilical cords). (b) Quantitation of luminal cross-sectional area at birth shows that the umbilical arteries are occluded whereas the veins remain patent (top) and have significantly thicker walls (bottom) (n = 20 cords, error bars indicate mean ± S.E.M., whiskers

*Figure 1 continued on next page*

Figure 1 continued

indicate minimum and maximum values. ***, p<0.001. (c) Alcian blue, eosin (pink) and nuclear fast red staining of umbilical vessel cross-sections shows a proteoglycan-rich (blue) inner TM in the umbilical artery but not the vein. Quantified staining intensity is shown on the right (n = 6 umbilical cords, whiskers indicate minimum and maximum values, ***, p<0.001). (d) Chondroitin sulfate (CS), heparan sulfate (HS), aggrecan and versican immunofluorescence (n = 4 cords for each antibody) showing that CS staining corresponds with aggrecan and versican staining and alcian blue in (c). (e) Volcano plots illustrating differential gene expression between human umbilical artery (red) and vein (green) (top, n = 4 umbilical arteries and veins) and differential gene expression between human umbilical artery inner TM (red) and the outer TM (green) (bottom, n = 2). (f) RNA in situ hybridization shows robust *ACAN* and *VCAN* expression (red signal) in the inner artery TM and weak expression in the vein (n = 3 umbilical cords for each in situ probe). * marks the vessel lumen. Brackets in c,d,f mark the TM. Wj, Wharton's jelly. Scale bars = 100 µm in c,d,f.

The online version of this article includes the following video and figure supplement(s) for figure 1:

**Figure supplement 1.** Morphological and cellular characteristics of the human umbilical arteries and veins at birth.
**Figure supplement 2.** Transcriptome comparison and pathway analysis of differences in the human umbilical artery and vein.
**Figure supplement 3.** Transcriptome comparison and pathway analysis of differences in the human inner umbilical artery tunica media (TM) vs the outer tunica media.
**Figure 1—video 1.** Synchrotron image stack of an umbilical artery.
https://elifesciences.org/articles/60683#fig1video1
**Figure 1—video 2.** Synchrotron image stack of an umbilical vein.
https://elifesciences.org/articles/60683#fig1video2

Neo-epitope antibodies detecting ADAMTS-cleaved aggrecan and versican (anti-NITEGE and anti-DPEAAE, respectively) (*Lark et al., 1995*; *Sandy et al., 1992*; *Sandy et al., 2001*) showed strong staining throughout the venous TM and in the outer arterial TM, but not the inner arterial TM (*Figure 2b*). Thus, proteoglycan accumulation in the inner TM of the umbilical artery may result from higher *ACAN* and *VCAN* expression and less proteolysis. In contrast, lower *ACAN* and *VCAN* expression and greater ADAMTS levels within the umbilical vein may preclude proteoglycan accumulation.

## ADAMTS-mediated differential proteoglycan abundance in the umbilical artery and vein is evolutionarily conserved

We postulated that abundant hydrated proteoglycans in the inner arterial TM provided compressive stiffness that could not only prevent kinking and premature occlusion but could potentially facilitate rapid umbilical artery closure at birth. If so, similar adaptations should be present in other mammals. Analysis of umbilical cords from nine large primate and non-primate mammals disclosed similar dimorphism, namely, umbilical arteries were occluded and had thicker walls with similar infolding of the inner arterial TM (*Figure 3a*) and strong Alcian blue and CS-staining, contrasting with veins (*Figure 3a,b*). Anti-aggrecan and anti-NITEGE stained several animal species, confirming aggrecan abundance in the inner arterial TM and robust aggrecan cleavage resulting from ADAMTS protease activity in the outer TM of the artery and the TM of the vein (*Figure 3c–d*, *Figure 3—figure supplement 1*).

## Aggrecan and ADAMTS1 are necessary for normal umbilical cord morphogenesis

Mouse umbilical cords also demonstrated vascular dimorphism (*Figure 4a*), suggesting that genetically modified mice would provide mechanistic insights into proteoglycan dynamics and its impact. Aggrecan and versican immunofluorescence showed strong staining in the mouse umbilical artery inner TM and adventitia, with weaker staining in the veins (*Figure 4b*). *Acan*, *Vcan* and *Adamts1,4,5,9* RNA-ISH at early (E12.5) and late (E18.5) gestational stages showed that *Acan* and *Vcan* were strongly expressed in the umbilical arteries (*Figure 4—figure supplement 1a*). *Adamts1* was the most highly expressed proteoglycanase in the mouse umbilical vein just prior to parturition (E18.5) (*Figure 4—figure supplement 1a*), evidenced by strong β-gal staining in venous TM, adventitia and endothelium; the inner umbilical artery TM and endothelium of *Adamts1*[lacZ/+] embryos lacked β-gal staining (*Figure 4c*). Although *Adamts9* mutant embryos were previously observed to have short umbilical cords, abnormal umbilical artery development, and to die by 14.5 days of gestation (*Nandadasa et al., 2015*), umbilical cord development was not previously investigated in

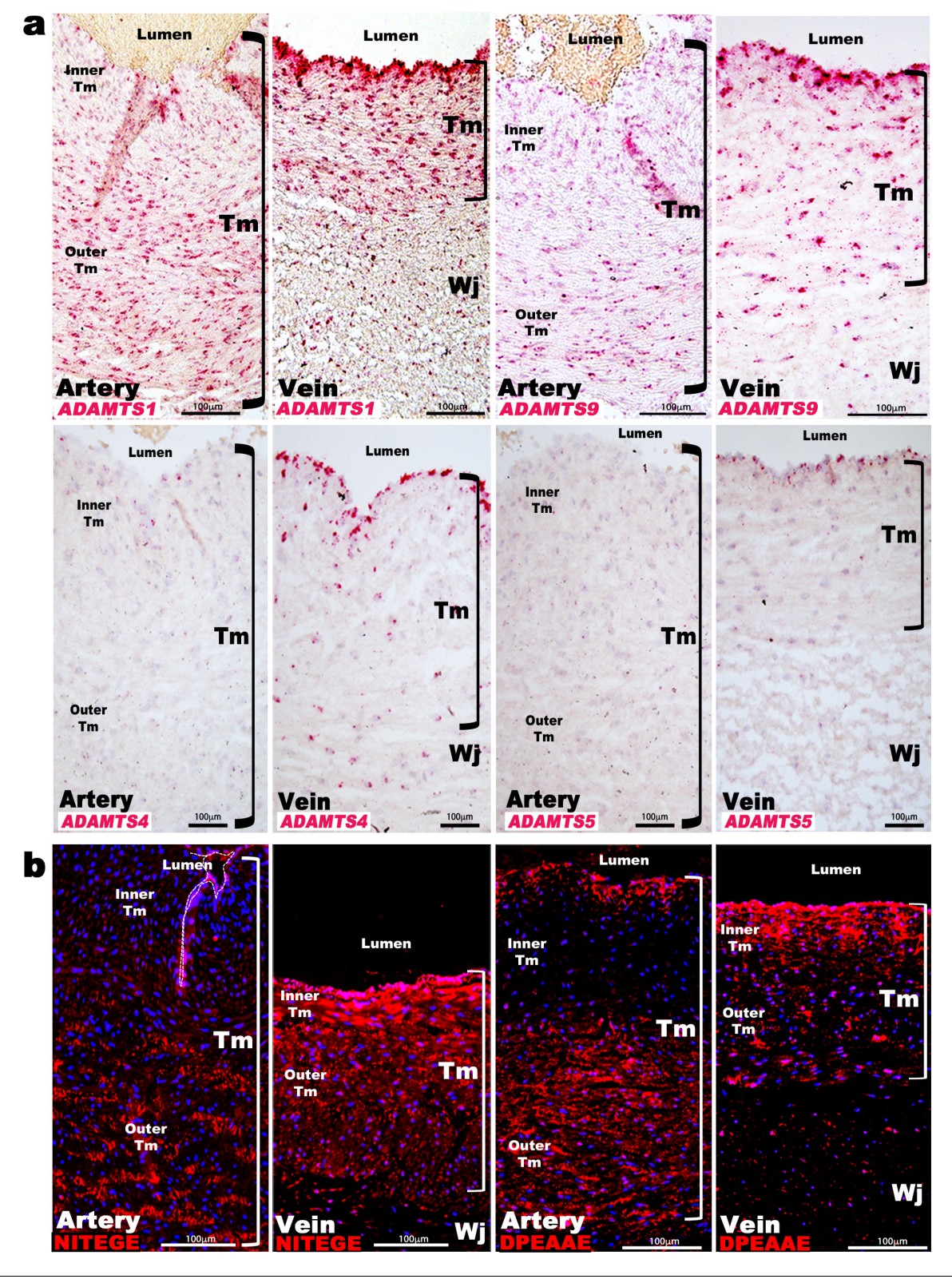

**Figure 2.** ADAMTS proteoglycanases are highly expressed and active in the human umbilical vein. (a) RNA in situ hybridization shows robust *ADAMTS1* and *ADAMTS9* expression in umbilical vein endothelium and tunica media (TM) and in outer arterial TM. Robust *ADAMTS4* and *ADAMTS5* expression was confined to the venous endothelium, with moderate ADAMTS4 expression and minimal ADAMTS5 expression in SMC (n = 3 umbilical cords for each probe). (b) ADAMTS-cleaved aggrecan (anti-NITEGE, red) and versican (anti-DPEAAE, red) both showed strong ADAMTS proteolytic activity

*Figure 2 continued on next page*

*Figure 2 continued*

throughout the venous wall and the outer artery TM. Unlike aggrecan, extensive versican proteolysis is seen in the arterial intima and sub-intima (n = 4 umbilical cords for each antibody). Wj, Wharton's jelly. The brackets mark TM boundaries. Scale bars in **a-b** = 100 μm.

mutants of the two genes implicated here as potentially critical for umbilical cord vascular dimorphism, *Acan* and *Adamts1*.

The *Adamts1*$^{-/-}$ mutant is an insertion of an IRES lacZ-bearing cassette into intron 1 of the gene (*Oller et al., 2017*). This insertion reveals *Adamts1* expression via staining for ß-galactosidase activity, and eliminated expression from the targeted allele but in its hemizygous state, the insertion led to reduction in both mRNA and protein (*Oller et al., 2017*). The *Acan*$^{cmd-Bc}$ allele is a spontaneous mutation found in a BALB/C colony (*Bell et al., 1986*) and resulted from deletion of exon 2 through exon 18 (*Krueger et al., 1999*). *Acan*$^{-/-}$ embryos do not survive past birth (*Krueger et al., 1999*; *Lauing et al., 2014*) and few surviving *Adamts1*$^{-/-}$ mice were identified at the time of weaning (*Oller et al., 2017*). *Acan* mutants are thought to succumb to respiratory failure resulting from soft tracheal cartilages and ribs, whereas the cause of *Adamts1*$^{-/-}$ lethality is unknown. At E18.5, *Acan*$^{-/-}$ and *Adamts1*$^{-/-}$ mutants each had significantly short umbilical cords (*Figure 4d*) demonstrating their requirement for proper umbilical cord development. Umbilical cord histology showed thinner vascular walls in *Acan*$^{-/-}$ umbilical vessels, and conversely, thicker vascular walls in *Adamts1*$^{-/-}$ umbilical vessels (*Figure 4e–g*). At earlier developmental stages (E12.5 to E14.5), lack of aggrecan did not affect either umbilical cord length or circumferential SMC reorientation (*Figure 4—figure supplement 1b–d*), which occurs around E13.5 and is defective in *Adamts9* mutants (*Nandadasa et al., 2015*). Furthermore, lack of aggrecan did not impair the survival of mouse embryos until parturition, since *Acan*$^{-/-}$ embryos were observed at the expected Mendelian ratio at E18.5 (*Figure 4—figure supplement 1c*). Thus, *Acan* and *Adamts1* appear to be involved in umbilical vessel development from early gestation, but their functions manifest near parturition.

## Contrasting SMC phenotypes in *Acan* and *Adamts1*-deficient umbilical cords

The arterial and venous lumina were smaller in *Adamts1*$^{-/-}$ mice relative to wild-type, indicating that their thicker vascular walls compromised luminal diameter, and larger in *Acan*$^{-/-}$ mice (*Figure 4e–g*). Phospho-histone H3 staining revealed fewer proliferating cells in *Acan*$^{-/-}$ umbilical cords at E18.5 (*Figure 4h*). Immunostaining for SMC markers smooth muscle α-actin (SMA), smooth muscle myosin heavy chain (SMMHC), and phosphorylated myosin light chain (pMLC) showed weaker intensity in *Acan*$^{-/-}$ umbilical arteries compared to wild-type (*Figure 5a,b*). In contrast, *Adamts1*$^{-/-}$ umbilical vessels showed stronger SMA, SMMHC and pMLC staining than wild-type littermates and apparent overgrowth of the arterial and venous walls (*Figure 5c*). Intriguingly, endomucin, a venous endothelium-specific marker (*dela Paz and D'Amore, 2009*), also stained *Adamts1*$^{-/-}$ umbilical arterial endothelium (*Figure 5c*) suggesting that ADAMTS1 may have a role in specifying artery/vein identity. Immunostaining of E17.5 *Adamts1*$^{-/-}$ umbilical cords indicated a crucial role for ADAMTS1 in regulating proteoglycan dynamics in the mouse umbilical cord. Specifically, we observed robust aggrecan and versican accumulation in the *Adamts1*$^{-/-}$ umbilical vein and in the outer TM of the *Adamts1*$^{-/-}$ umbilical artery (*Figure 6a–d*) with severe reduction of aggrecan and versican neo-epitope staining (*Figure 6a–d*).

## Differential SMC contraction in the bilayered umbilical arteries

Despite uniform staining with SMC markers in human umbilical vascular SMC, co-staining with serine[20]-phosphorylated myosin light chain (pMLC) marking contractile SMCs (*Dougherty et al., 2014*) revealed that human umbilical arteries had more contractile SMCs than the vein, predominantly in the outer TM (*Figure 7a,b*). This suggests that outer umbilical artery SMCs are principally responsive to vasoconstriction stimuli at birth, whereas inner SMCs are relatively non-contractile. We hypothesized that an outer ring of contracting SMCs could drive the CSPG-rich inner arterial TM centripetally, occluding the lumen, and addressed this possibility initially using ex vivo biomechanical testing of late-gestation mouse umbilical vessels (*Figure 7c*, *Figure 7—figure supplement 1*). Mouse umbilical arteries had a smaller lumen, as expected at E18.5, and deformed less when loaded

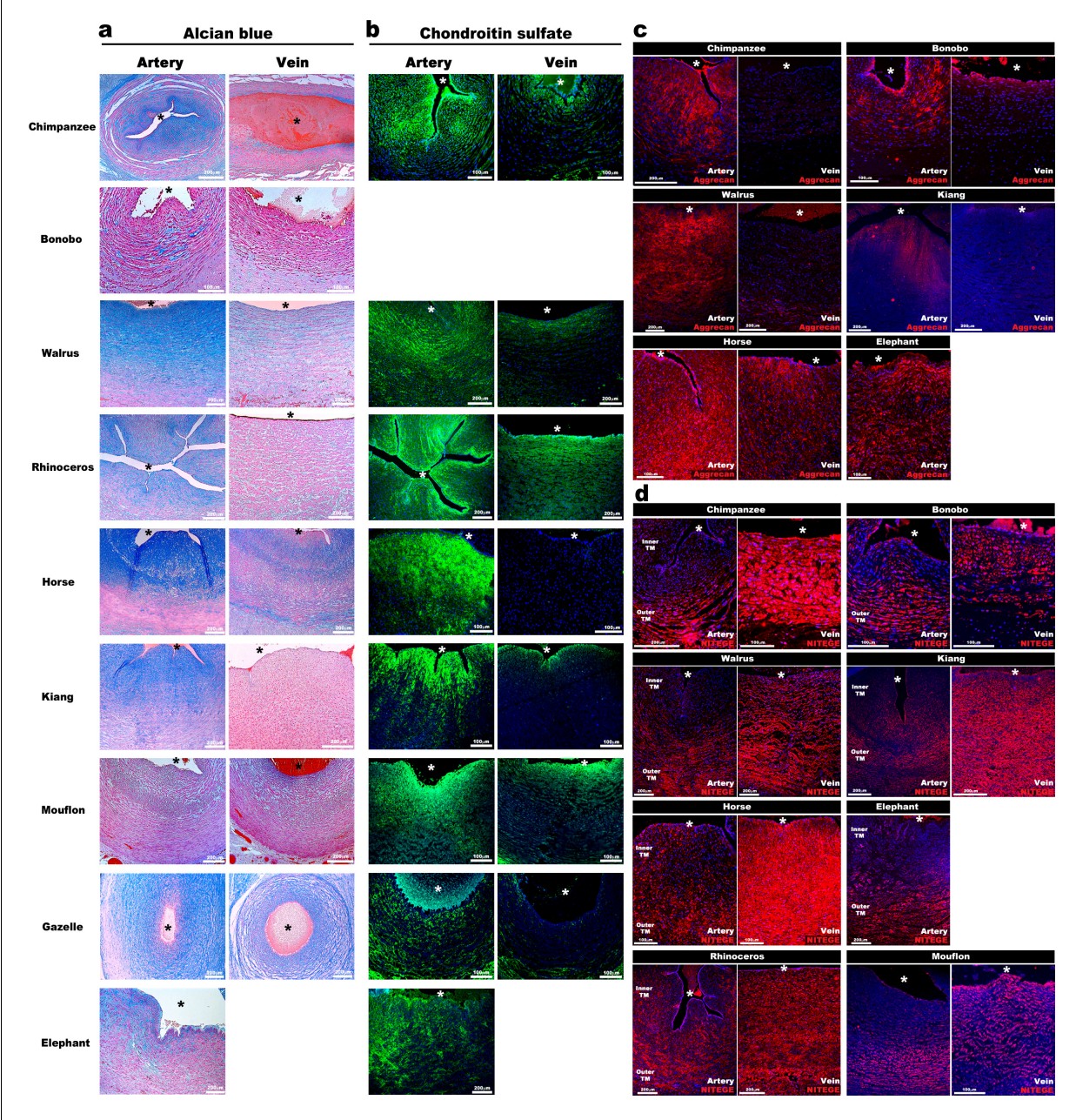

**Figure 3.** Aggrecan enrichment in the inner umbilical artery tunica media (TM) and its proteolysis in the umbilical vein is a characteristic of large mammals. (a) Alcian blue-eosin staining of umbilical cord sections shows proteoglycan enrichment (blue) in the inner arterial tunica media (TM). The elephant umbilical vein was unavailable. (b) Anti-CS immunofluorescence (7D4, green) shows enrichment in the inner arterial TM. Bonobo cords lacked 7D4 reactivity. (c,d) Aggrecan and anti-NITEGE immunostaining from reactive species showed aggrecan enrichment in the inner arterial TM and aggrecan proteolysis in the vein and outer artery TM. n = 3 for Gazelle and n = 1 for other mammals. Triplicate sections were stained from each animal cord. Scale bars = 100 µm and 200 µm. * indicates the vessel lumen.

The online version of this article includes the following figure supplement(s) for figure 3:

**Figure supplement 1.** Aggrecan cleavage site conservation in mammals.

mechanically, namely, they exhibited lower (circumferential) distensibility and especially (axial) extensibility under passive conditions compared to the umbilical veins (*Figure 7—figure supplement 1a–d*). Umbilical arteries constricted significantly (30–50% reduction in measured outer diameter at 25 mm Hg fixed pressure), causing complete luminal occlusion verified by optical coherence (OCT)

imaging, which was not observed in the umbilical veins (*Figure 7c*). Cross-sectional area measurements at fixed lengths revealed wall volume reductions during vasoconstriction (*Figure 7—figure supplement 1e*), less in the umbilical vein (~35%) than the umbilical artery (~50%), suggesting fluid exudation from the wall under forceful SMC contraction.

## Computational modeling of arterial occlusion

These biomechanical tests of mouse umbilical cords, together with histological and immunostaining findings from human cords, motivated and informed a novel computational model of the umbilical artery incorporating its complex bilayered, multi-constituent structure (GAG-rich inner layer and contractile SMC-rich outer layer; *Figure 7d*) and multiaxial mechanical loading: axial extension, luminal pressurization, active contraction by SMCs, and intramural swelling of the inner layer that regulates tissue volume locally based on GAG content. Nonlinear regression of biaxial mechanical data from passive tests of the murine vessels identified best-fit values of the material parameters in the baseline constitutive model, while data from active contraction studies guided the selection of the associated active constitutive parameters (*Table 1*). Model-based parametric studies examined combinations of different levels of GAG-driven swelling and SMC-generated active stress to identify their roles in umbilical artery closure at different levels of fixed luminal pressure. Increasing inner layer swelling in the absence of active outer layer stress narrowed the lumen at a fixed pressure, as expected given the constraining effect of the outer stiff passive matrix (*Figure 7—figure supplement 2a*). This trend reversed in the presence of active stresses, with increasing inner layer GAGs able to oppose vasoconstriction if overall wall volume remained constant (*Figure 7—figure supplement 2b*). Thus, increased inner layer swelling attenuates the ability of SMC contraction to prematurely reduce luminal radius, as revealed by varying the active stress parametrically for different fixed values of inner layer swelling. Importantly, the model predicted a sharp transition from a widely patent to a narrowed lumen due to small changes in active stress at lower values of swelling whereas radius changes were more gradual at higher values of swelling (*Figure 7—figure supplement 2b*). This transition, at which a decrease in volume of the inner layer associates with larger or smaller luminal radii for values of active stress below or above $T_{act} \cong 50$ kPa (a key parameter of active stress generation) appears to be close to the in vivo value. Hence, consistent with ex vivo findings (*Figure 7—figure supplement 1*), it appears that inner layer volume loss during increased SMC contraction aids vessel narrowing. Regardless, the inner radius reached nearly constant values for increasing levels of active stress (*Figure 7—figure supplement 2b*). Thus, contraction alone is insufficient to occlude the vessel.

## Folding of the arterial inner layer is necessary for vascular occlusion

Given the consistent histological finding of inner arterial TM infolding following birth, we modeled the biomechanics of superimposed inner layer buckling in the bilayered arterial model. Buckling can release energy stored in the inner layer during vasoconstriction-induced compression, thereby reducing the structural stiffness and resistance to SMC contraction. This analysis parametrically considered possible perturbations to the cylindrical geometry achieved at various levels of fixed luminal pressure and different values of swelling and actively generated wall stress. Examining the influence of the number of inner layer folds for different values of swelling disclosed higher inward buckling probability with more folds (*Figure 7d*). Since $T_{act}$ needed to cause buckling tended to plateau at 7 folds, we used 7 folds subsequently for illustrative purposes. $T_{act}$ needed to cause buckling decreased for a more swollen inner layer *Figure (7dFigure 7—figure supplement 2c*) and increased exponentially with inner layer volume loss. This finding was likely due to the less negative values of circumferential wall stress in the inner layer occurring with shrinkage (*Figure 7—figure supplement 2d*). We found that an arterial wall consisting solely of SMCs and uniform matrix maintained a mean positive circumferential stress in the inner layer during contraction, that prevented buckling. Thus, a delicate biomechanical balance exists – reduced inner layer volume allows a smaller radius to be achieved via SMC contraction prior to buckling (*Figure 7—figure supplement 2b*), thus aiding closure, yet excess volume reduction of the inner layer increases the active stress requirement for buckling and achieving complete vessel closure (*Figure 7d*). The umbilical artery can reduce its cylindrical radius dramatically at $T_{act}$ near a basal value of ~50 kPa, progressing to buckling and closure via a subsequent near-maximal contraction (*Figure 7e*). In agreement with the computational modeling,

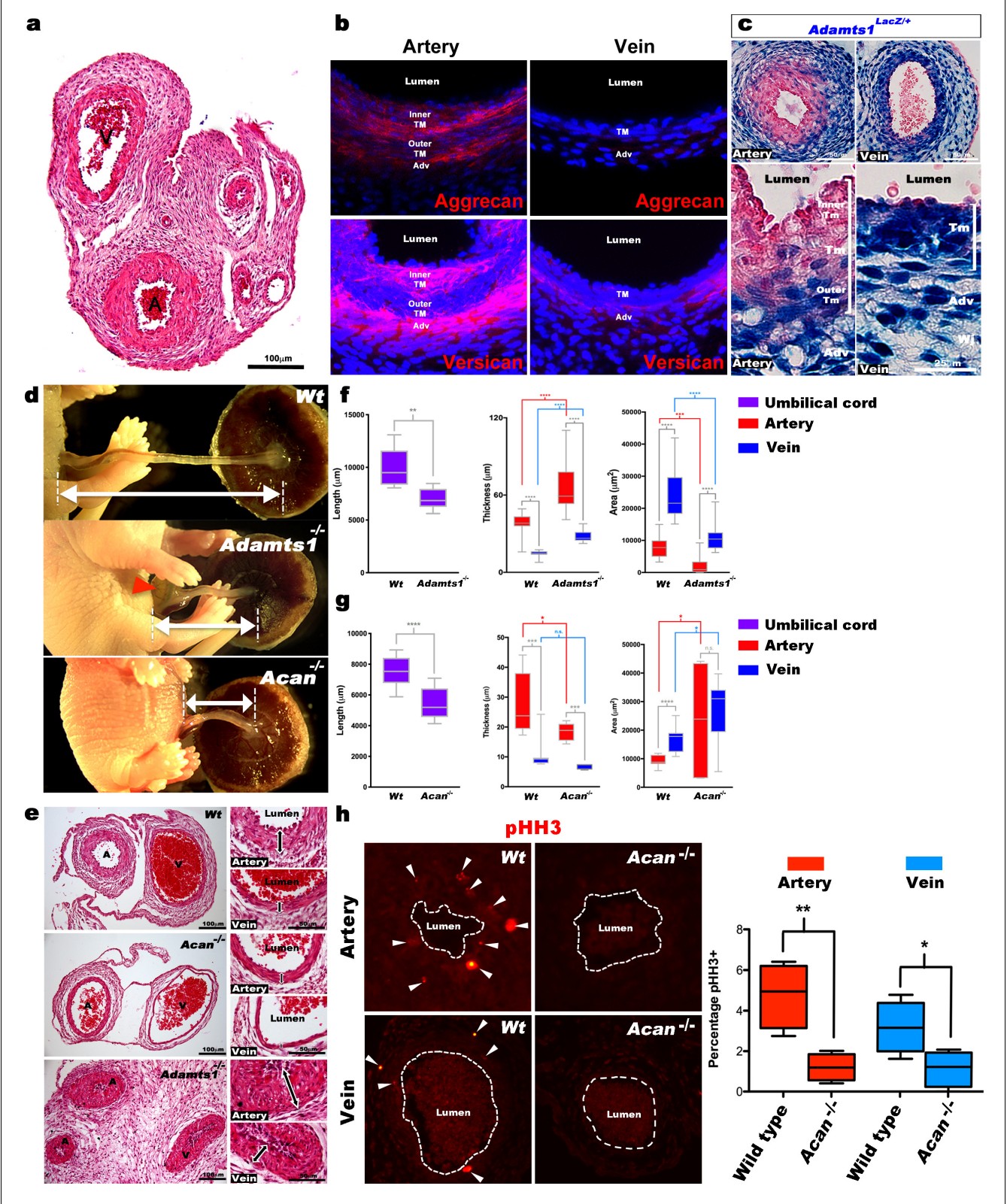

**Figure 4.** Defective morphogenesis in *Acan* and *Adamts1* mutant mouse umbilical cords. (**a**) H and E staining of E18.5 wild-type cords showing thicker umbilical arterial (A) and thinner venous (V) wall (n = 6 umbilical cords). (**b**) Aggrecan and versican localization (red, DAPI counterstain blue) in E18.5 wild-type cords showing staining in the arterial inner tunica media (TM) and adventitia but not the vein (n = 3 umbilical cords). (**c**) β-gal (blue) and eosin (red) staining of E18.5 *Adamts1^LacZ/+^* (*Adamts1^+/-^*) cord showing strong *Adamts1* expression in venous endothelium and TM and outer artery TM (n = 3

*Figure 4 continued on next page*

*Figure 4 continued*

umbilical cords). (**d**) Short umbilical cords in E18.5 *Adamts1*−/− and *Acan*−/− embryos compared to wild type. Red arrowhead indicates an omphalocele in *Adamts1*−/− embryos. (**e**) H & E staining of E18.5 wild type, *Acan*−/− and *Adamts1*−/− cord cross-sections showing thinner walls in *Acan*−/− umbilical vessels and thicker walls in *Adamts1*−/− umbilical vessels. (**f–g**) Cord length, TM thickness and vessel luminal area quantifications for *Adamts1*−/− (**f**) and *Acan*−/− mice (**g**) at E18.5 compared to wild-type littermates. *Acan*−/− umbilical cords show larger lumens and *Adamts1*−/− vessels show smaller lumens in (n = 7–11 umbilical cords each, whiskers indicate minimum and maximum values, *, p<0.05; **, p<0.01; ***, p<0.001; ****, p<0.0001). (**h**) Phospho-histone H3 (pHH3) staining shows significantly fewer proliferating cells (white arrowheads) in *Acan*−/− umbilical vessels. Dotted white lines mark the boundaries of vessel lumens (n = 4 cords each, whiskers indicate minimum and maximum values, **, p<0.001; *, p<0.05). Scale bars = 100 μm in (**a**), 25 μm in (**c**), 100 μm and 50 μm in (**e**).

The online version of this article includes the following figure supplement(s) for figure 4:

**Figure supplement 1.** ADAMTS, *Vcan* and *Acan* expression and impact of aggrecan loss on early mouse umbilical cord and vessel development.

20/25 of human umbilical arteries had 4 or more buckles, whereas those with no buckles in the area analyzed by histology were patent (*Figure 7f*). Other large mammalian species analyzed showed a similar phenomenon (*Figure 7f*). Thus, buckling of the proteoglycan-rich inner tunica media may be a crucial and evolutionarily conserved mechanism employed by all mammals for rapid arterial occlusion at birth.

## Discussion

We report two distinct umbilical cord blood vessel specializations that may facilitate rapid umbilical artery occlusion at birth: a distinct proteoglycan-rich inner arterial TM, generating a bilayered arterial wall, and selective contraction of SMCs in the outer layer (*Figure 8*). The rounded SMCs of the inner TM may be specialized for CSPG production rather than contraction, consistent with nuclear Sox9 staining, a function they exert prior to delivery. During delivery, lack of pMLC staining suggests that despite differentiated SMC marker expression, the inner cells play a passive role. Biomechanical testing and computational analysis confirm that selective proteoglycan enrichment in the inner arterial TM ensures that contracting SMCs in the outer TM can effectively occlude the arterial lumen at birth (*Figure 8*). Histologic and computational analysis showed buckling of the inner TM and fluid redirection into the resulting TM protrusions as critical mechanisms resulting from specialization of the inner and outer arterial TM. By in silico simulations of umbilical arteries with modulation of the contractile outer layer and proteoglycan-rich inner core, we demonstrate that complete occlusion can be achieved.

Our work suggests that a principal mechanism governing umbilical cord vascular dimorphism resides in extracellular matrix, namely, differentially regulated dynamics of aggrecan and versican, which may then modulate SMC development and differentiation. Other mammalian umbilical vessels examined, from animals as large as the walrus and elephant to as small as the mouse, showed similar CSPG and aggrecan modulation as in humans. Although *Vcan* mutant mice die before umbilical cord development is completed (*Yamamura et al., 1997*) and could not be studied, *Acan* and *Adamts1* mutants demonstrated their mechanistic contributions to the observed dimorphism. Our emphasis on aggrecan in the inner layer, with its abundant GAGs and their high fixed charge density, was relevant to the computational findings of the importance of inner layer swelling and buckling in response to SMC contraction. The umbilical vein contains fewer contractile SMCs and has scant aggrecan and versican. Hence, SMC activation in the umbilical vein does not occlude the lumen to the same degree as in the arteries, a contention supported by computational analysis. Given the evolutionary pressure to achieve hemostasis urgently in the artery rather than the vein, these findings suggest a highly evolved mechanism for preventing exsanguination of the newborn that is potentially relevant to other embryonic shunts that occlude rapidly at birth.

The computational model was built on a long history of studying murine arteries and veins (*Ferruzzi et al., 2013*), but was specialized to the GAG-rich inner layer and SMC-rich outer layer of the umbilical artery. Modeling the dynamics of associated volume changes would have required a mixture or poroelastic model and significantly more experimental data, including measurement of layer-specific permeabilities and fixed-charge densities. Instead, we modeled the quasi-equilibrated states using a well-accepted approach wherein the degree of swelling can be adjusted for each simulation (*Demirkoparan and Pence, 2007*; *Szafron et al., 2017*). Interestingly, prior results by others

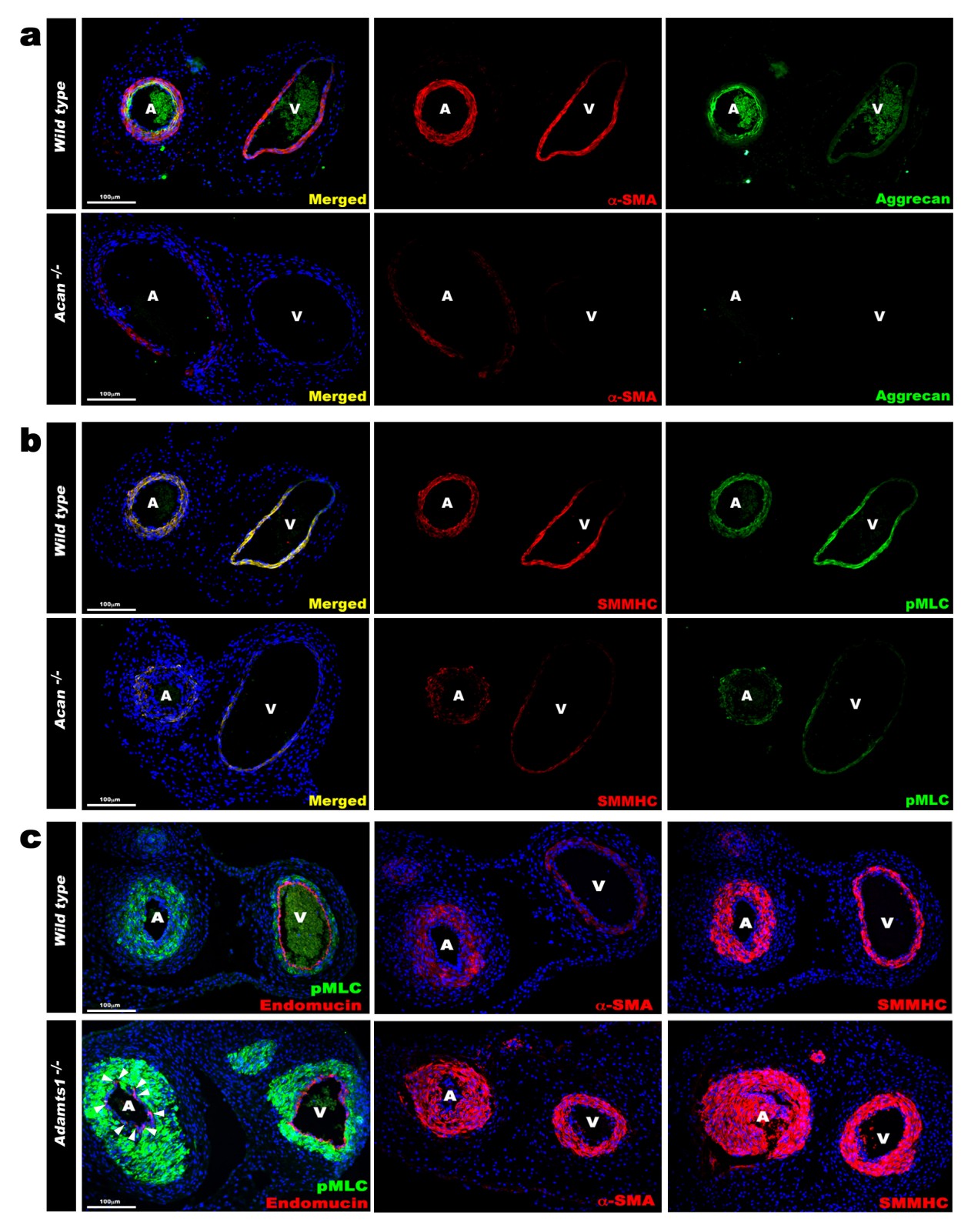

**Figure 5.** Contrasting smooth muscle cell (SMC) phenotype modulation in *Acan* and *Adamts1*-deficient umbilical vessels. (**a**) Aggrecan (green) and α-SMA staining (red) in E18.5 umbilical cords show loss of aggrecan and weak α-SMA staining in *Acan*⁻ᐟ⁻ vessels (n = 3 umbilical cords each genotype). (**b**) Smooth muscle myosin heavy chain (SMMHC, red) and phosphorylated myosin light chain (pMLC, green) staining in E18.5 umbilical cords showing dramatic signal attenuation in the *Acan*⁻ᐟ⁻ vessels (n = 3 umbilical cords each genotype) (**c**) pMLC (green), endomucin (red), α-SMA (red, center panels)

*Figure 5 continued on next page*

*Figure 5 continued*

and SMMHC (red, right-hand panels) staining shows blunted dimorphism of *Adamts1*[-/-] umbilical artery and vein with stronger expression of differentiated SMC markers in *Adamts1*[-/-] umbilical vessels and acquisition of endomucin, a venous endothelium marker, by arterial endothelium (n = 3 umbilical cords each genotype) Scale bars = 100 μm in (a–c).

showed that swelling of an initially unloaded, unilayered cylindrical tube consisting of a neo-Hookean material (which we used to model GAG-rich tissue) increases luminal diameter if the tube is unconstrained (*Demirkoparan and Pence, 2007*). The tube must be constrained, such as by a stiff outer layer surrounding the swollen layer if swelling is to decrease luminal diameter (*Szafron et al., 2017*). The abundance of contractile SMCs in the stiffer outer layer and their basal tone may in fact enhance outer layer stiffness contributed by extracellular matrix and hence buckling appears to be essential to augment contraction-induced closure of the umbilical artery (cf [*Moulton and Goriely, 2011*]).

This observed architecture of the umbilical cord is likely to have supported survival of mammalian species, humans included, well before formal obstetric involvement in labor. Cord clamping is the default practice today and has the sanction of convention, offering the option of immediate neonatal resuscitation if needed. In regard to the umbilical artery, it would replicate the effect of a natural and apparently conserved physiologic process that interrupts its blood flow during transition from fetal to neonatal life. The latest recommendation to clamp the cord later rather than immediately after birth appears to align better with the delayed closure of the umbilical vein. The present studies

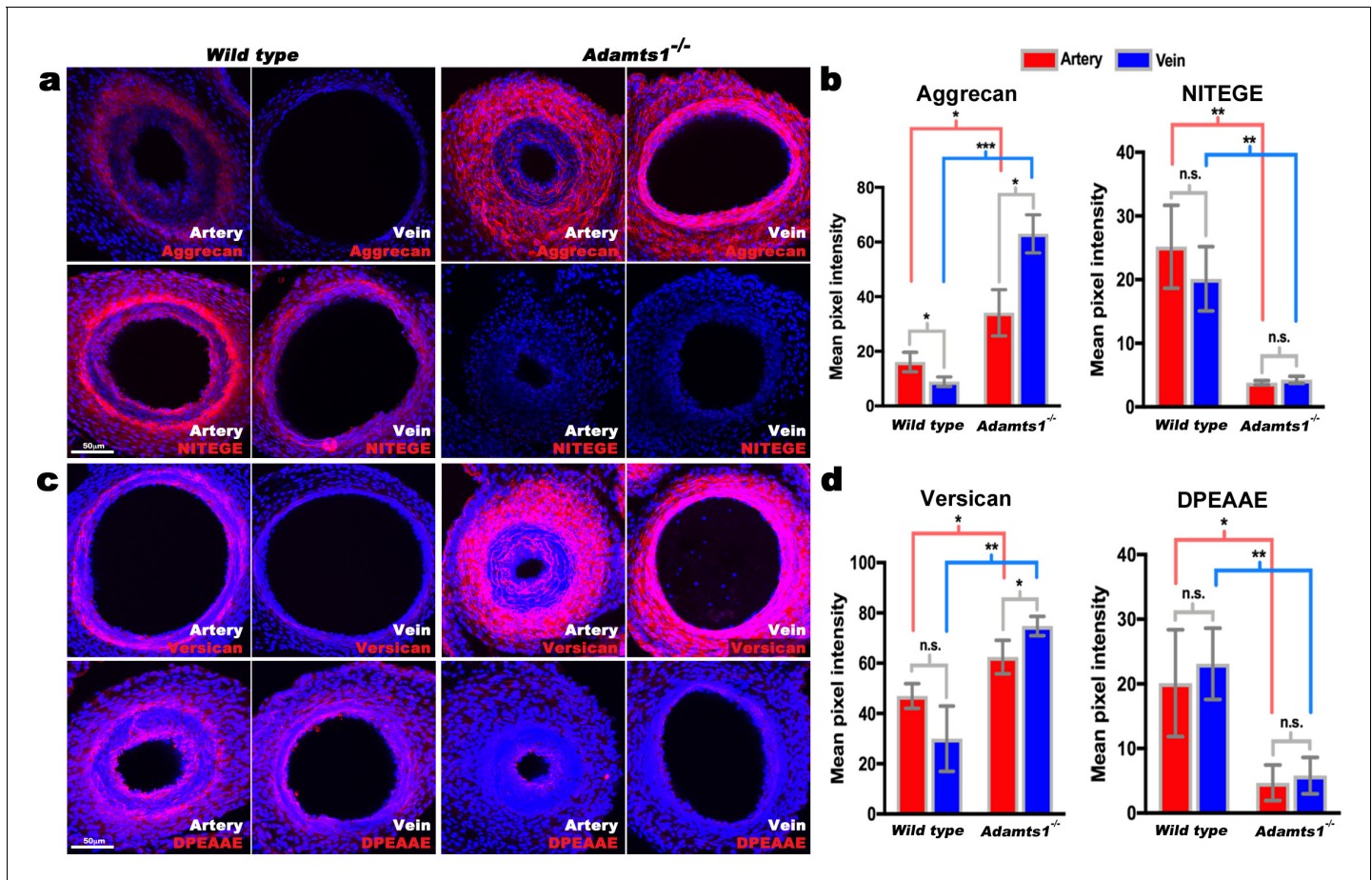

**Figure 6.** Reduced aggrecan and versican proteolysis in *Adamts1*[-/-] umbilical vessels. (a,b) E17.5 *Adamts1*[-/-] umbilical vessels show increased aggrecan staining and reduced anti-NITEGE staining in (a), quantified in (b) (n = 3 cords each genotype, error bars indicate mean ±S.D.*, p<0.05; **, p<0.01; ***, p<0.001). (c,d) *Adamts1*[-/-] umbilical vessels show increased versican (c) and reduced anti-DPEAAE staining quantified in (d) (n = 3 cords each genotype, error bars indicate mean ±S.D. *, p<0.05; **, p<0.01). Scale bars = 50 μm in (a–c).

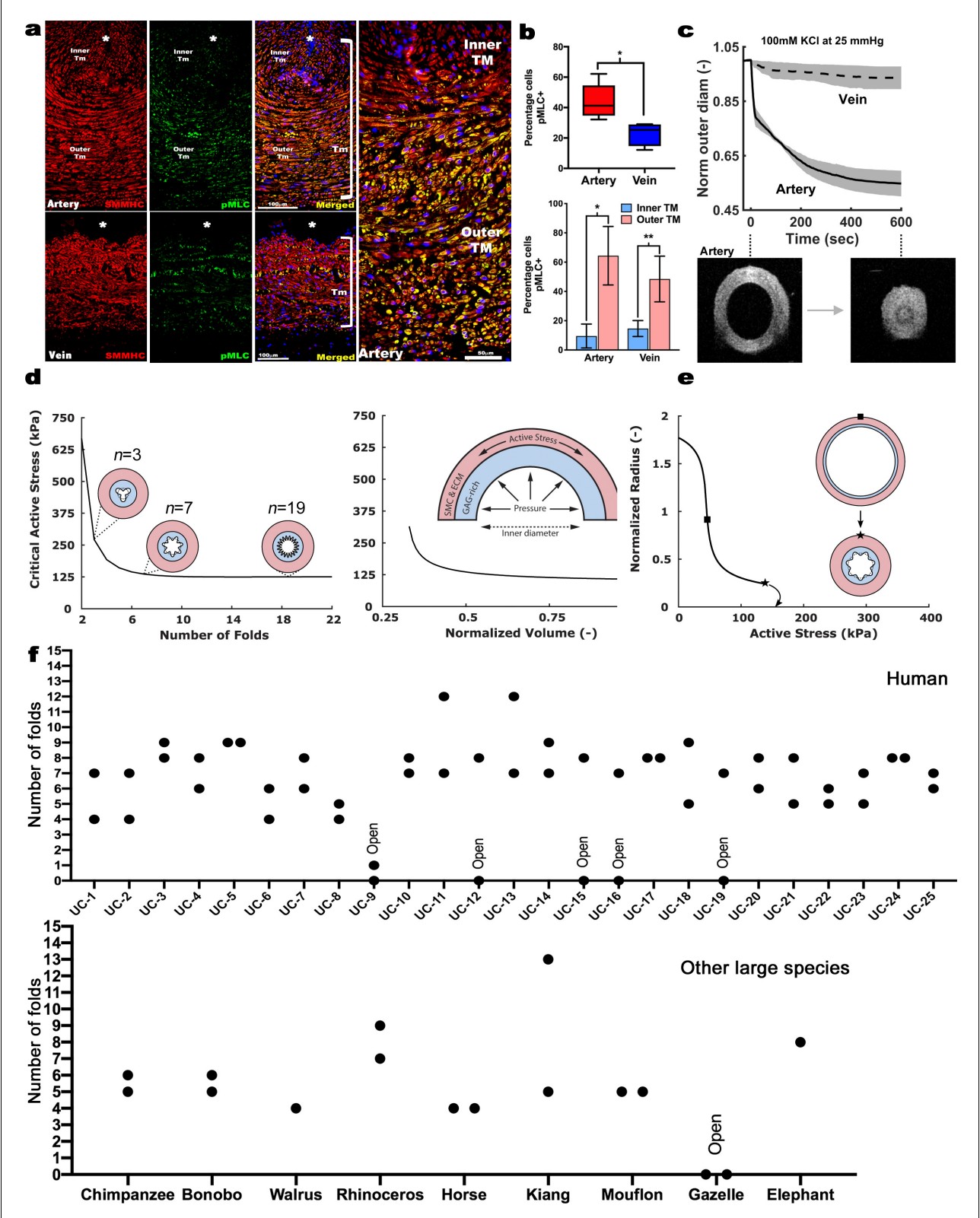

**Figure 7.** Contraction-induced buckling ensures effective closure of the umbilical artery at birth. (**a**) Smooth muscle myosin heavy chain (SMMHC-red) and serine-20 phosphorylated myosin light-chain (pMLC-green) show more contraction-primed SMCs in the outer arterial tunica media (TM, white brackets) than the umbilical vein. Scale bars are 100 μm. (**b**) Quantitation of pMLC$^+$ SMCs in the artery (red) and vein (blue) (top, n = 5 arteries, four veins, whiskers indicate minimum and maximum values, *, p<0.05) and inner and outer TM of both reveal similar distributions but more pMLC$^+$ SMC in
*Figure 7 continued on next page*

*Figure 7 continued*

the outer artery TM (bottom, n = 3 arteries, four veins, error bars indicate mean ±S.D. *, p<0.05; **, p<0.01). (**c**) Differential contraction of murine umbilical artery and vein stimulated by 100 mM potassium chloride (KCl) under biaxial loading confirms greater contractility in the artery, with OCT images prior to and following contraction-induced arterial closure (n = 4 arteries and n = 4 veins). (**d**) Computational simulations of a bilayered artery with contractile SMCs in the outer layer and swollen inner layer: critical contractile stress values leading to buckling for (*left*) different numbers of folds for a normalized inner layer volume of 0.5 and (*right)* decreasing values of normalized volume of the inner layer for seven folds. (**e**) Normalized inner radius as a function of contractile stress for inner layer volume change of 0.5 and 7 folds. The states for the inflection (square) and critical active stress (star) are illustrated by the schematics; complete closure achieved with contraction-induced buckling. All simulations were run for 25 mmHg pressure. Due to the linear stability analysis, the amplitude of the folds in the buckled schematics is illustrative. (**f**) Number of buckles observed in human (top, indicated as umbilical cord (UC)1–25) and other large mammalian (bottom) umbilical arteries. Both arteries per cord were included. Open vessel lumens are indicated where observed.

The online version of this article includes the following figure supplement(s) for figure 7:

**Figure supplement 1.** Differential biomechanical properties of arteries and veins of the mouse umbilical cord.
**Figure supplement 2.** Computational results for a model umbilical artery.

in humans and other mammals show that evolution has devised an umbilical cord-intrinsic mechanism that facilitates rapid arterial occlusion at birth, leaving the vein patent and permitting a placental infusion. This unfailing sequence ensures continuation of mammalian species without other formal intervention, since evolutionary success is not about minimizing poor outcomes, but ensuring survival of a significant majority.

# Materials and methods

### Key resources table

| Reagent type (species) or resource | Designation | Source or reference | Identifiers | Additional information |
|---|---|---|---|---|
| Gene (*Homo sapiens*) | *ACAN* | GenBank | RRID:HGNC:319 | Chondroitin sulphate proteoglycan 1 |
| Gene (*Homo sapiens*) | *VCAN* | GenBank | RRID:HGNC:2464 | Chondroitin sulphate proteoglycan 2 |
| Gene (*Homo sapiens*) | *ADAMTS1* | GenBank | RRID:HGNC:217 | ADAM metallopeptidase with thrombospondin type 1 motif 1 |
| Gene (*Homo sapiens*) | *ADAMTS4* | GenBank | RRID:HGNC:220 | ADAM metallopeptidase with thrombospondin type 1 motif 4 |
| Gene (*Homo sapiens*) | *ADAMTS5* | GenBank | RRID:HGNC:221 | ADAM metallopeptidase with thrombospondin type 1 motif 5 |
| Gene (*Homo sapiens*) | *ADAMTS9* | GenBank | RRID:HGNC:13202 | ADAM metallopeptidase with thrombospondin type 1 motif 9 |
| Genetic reagent (*Mus musculus*) | *Acan*cmd-Bc (C57BL/6J background) | **Krueger et al., 1999** | RRID:MGI:1855999 | *Acan* null allele |
| Genetic reagent (*Mus musculus*) | *Adamts1*tm1Dgen (C57BL/6J background) | **Oller et al., 2017** | RRID:MGI:5427602 | *Adamts1* null and *LacZ* reporter allele |
| Antibody | Mouse monoclonal smooth muscle α-actin (α-SMA) Cy3 conjugated | Millipore Sigma C6198 | RRID:AB_476856 | IF (1:400) |
| Antibody | Rat monoclonal smooth muscle myosin heavy chain (SMMHC) | Kamiya Biomedical MC352 | RRID:AB_1241986 | IF (1:400) |
| Antibody | Rabbit polyclonal Serine-20 phosphorylated myosin light chain (pMLC) | Abcam Ab2480 | RRID:AB_303094 | IF (1:200) |

*Continued on next page*

*Continued*

| Reagent type (species) or resource | Designation | Source or reference | Identifiers | Additional information |
|---|---|---|---|---|
| Antibody | Rabbit polyclonal anti-serine-10 phosphorylated histone H3 (pHH3) | Millipore Sigma 06–570 | RRID:AB_310177 | IF (1:200) |
| Antibody | Mouse monoclonal anti-chondroitin sulfate (7D4) antibody | Bruce Caterson/Clare Hughes laboratory (*Sorrell et al., 1990*) | RRID:AB_2864328 | IF (1:200) |
| Antibody | Mouse monoclonal FITC-conjugated anti-heparan sulfate (10E4) antibody | US Biological H-1890 | RRID:AB_10013601 | IF (1:200) |
| Antibody | Rabbit polyclonal Anti-versican (pVC) | Apte laboratory (*Foulcer et al., 2014*) | RRID:AB_2864327 | IF (1:400) human tissue |
| Antibody | Rabbit polyclonal anti-versican GAG-beta | Millipore Sigma AB1033 | RRID:AB_90462 | IF (1:400) mouse tissue |
| Antibody | Rabbit polyclonal anti-versican V0/V1 neo epitope DPEAAE | Invitrogen PA1-1748A | RRID:AB_2304324 | IF (1:200) human/mouse |
| Antibody | Rabbit polyclonal anti-aggrecan | Millipore Sigma AB1031 | RRID:AB_90460 | IF (1:400) all species |
| Antibody | Rabbit polyclonal anti-aggrecan neo epitope NITEGE | Invitrogen PA1-1746 | RRID:AB_2242021 | IF (1:200) all species |
| Antibody | Rat monoclonal anti-endomucin antibody (clone eBioV.7C7) | Invitrogen 14-5851-85 | RRID:AB_891531 | IF (1:400) |
| Antibody | Rabbit polyclonal anti-SOX9 antibody | Millipore Sigma AB5535 | RRID:AB_2239761 | IF (1:200) |
| Commercial assay or kit | *ACAN* RNAscope In situ probe | ACD bio | 506841 | Human probe |
| Commercial assay or kit | *Acan* RNAscope In situ probe | ACD bio | 439101 | Mouse probe |
| Commercial assay or kit | *VCAN*-E8 RNAscope In situ probe | ACD bio | 452241 | Human probe detects exon 8 |
| Commercial assay or kit | *Vcan*-E8 RNAscope In situ probe | ACD bio | 428321 | Mouse probe detects exon 7 |
| Commercial assay or kit | *ADAMTS1* RNAscope In situ probe | ACD bio | 524501 | Human probe |
| Commercial assay or kit | *Adamts1* RNAscope In situ probe | ACD bio | 463361 | Mouse probe |
| Commercial assay or kit | *ADAMTS4* RNAscope In situ probe | ACD bio | 537341 | Human probe |
| Commercial assay or kit | *Adamts4* RNAscope In situ probe | ACD bio | 497161 | Mouse probe |
| Commercial assay or kit | *ADAMTS5* RNAscope In situ probe | ACD bio | 427611 | Human probe |
| Commercial assay or kit | *Adamts5* RNAscope In situ probe | ACD bio | 427621 | Mouse probe |
| Commercial assay or kit | *ADAMTS9* RNAscope In situ probe | ACD bio | 445321 | Human probe |
| Commercial assay or kit | *Adamts9* RNAscope In situ probe | ACD bio | 400441 | Mouse probe |
| Commercial assay or kit | RNAscope 2.5 HD Red In situ detection kit | ACD bio | 322350 | Used for detecting all probes in this study |

*Continued on next page*

*Continued*

| Reagent type (species) or resource | Designation | Source or reference | Identifiers | Additional information |
|---|---|---|---|---|
| Software, algorithm | Affymetrix Transcriptome Analysis Console, RMA-SST sketch algorithm | Affymetrix TAC 4.0 | RRID:SCR_018718 | Used for gene expression analysis for all microarray experiments in the study |
| Software, algorithm | R | Bell Laboratories/R Foundation for Statistical Computing Ver. 3.5.2. | RRID:SCR_001905 | Used for statistical computing of microarray data |
| Software, algorithm | GraphPad Prism | GraphPad | RRID:SCR_002798 | Used for statistical computing of other experimental data |

Abbreviations, IF, Immunofluorescence.

## Human and large mammal cords

Twenty-five human umbilical cords were obtained from uncomplicated term pregnancies either after vaginal birth (n = 13) or Cesarean section for obstetric indications (n = 12), that is, malpresentation or repeat Cesarean section. The samples were collected under an IRB exemption from Cleveland Clinic (EX-0118) for use of discarded tissue without patient identifiers. These cords were used for histological/immunohistologic analysis, synchrotron imaging, RNA in situ hybridization, and transcriptomics of inner versus outer umbilical artery TM. Animal cord sections were provided by Disease Investigations, Institute for Conservation Research, San Diego Zoo Global from the Benirschke archive.

## Mutant mice

The *Adamts1* transgenic allele (*Adamts1*[tm1Dgen]), referred to herein as *Adamts1*[-/-], was produced by insertion of an IRES-lacZ cassette into intron 1 of *Adamts1* using homologous recombination in mouse embryonic stem cells (**Oller et al., 2017**). The *Acan*[cmd-Bc] allele was previously described (**Krueger et al., 1999**) and is referred to herein as *Acan*[-/-]. Mice were handled under standard conditions under approved IACUC protocols at the Cleveland Clinic (IACUC protocol nos. 18–1996 and 18–2045) and University of Chicago (IACUC protocol no. 43751). Mutant mouse embryos were collected by timed matings of heterozygous mice by the detection of copulation plugs (taken as day 0.5 of gestation). Embryos were dissected out immediately following $CO_2$ mediated euthanasia and cervical dislocation of pregnant mice. Dissected whole embryos, with umbilical cords and placentas attached, were fixed in 4% paraformaldehyde at 4°C overnight. Umbilical cords were dissected out the following day and washed thrice in PBS and embedded in paraffin or in 4% agarose for vibratome sectioning as previously described (**Nandadasa et al., 2015**).

**Table 1.** Model parameters fixed for all simulations of the umbilical artery, determined primarily from the biaxial biomechanical data and histological findings.

| Parameters | Description | Values |
|---|---|---|
| $A$, $B$, $C$ | Unloaded inner, interface, outer radius | 161.77, 206.86, 236.92 μm |
| $\lambda_z$ | Loaded axial stretch | 1.28 |
| $\mu_1$, $\mu_2$ | GAG/matrix shear modulus inner, outer layer | 3.0 kPa, 0.1 kPa |
| $c_1^1, c_2^1$ | Axial fiber family material parameters | 0.013 kPa, 11.65 |
| $c_1^2, c_2^2$ | Circumferential fiber family material parameters | 2.66 kPa, 1.20 |
| $c_1^{3,4}, c_2^{3,4}$ | Diagonal fiber families' material parameters | 3.04 kPa, 4.23 |
| $\eta^3, \eta^4$ | Diagonal fiber families' alignment parameter | 41.92°, −41.92° |
| $\lambda_m, \lambda_0$ | Maximum, minimum contractile stretch | 2.5, 0.2 |

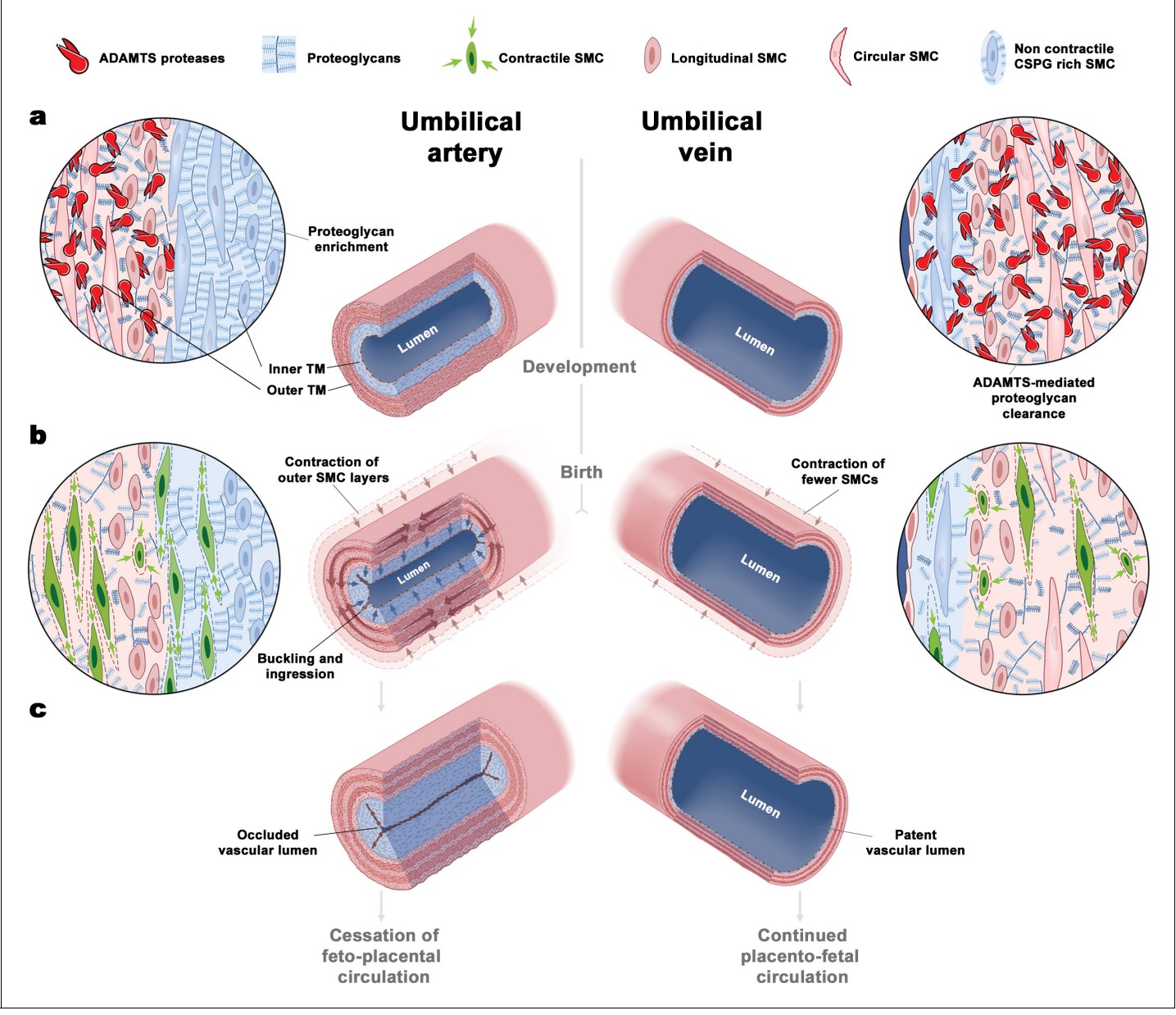

**Figure 8.** The unique bilayered design of the umbilical artery underlies its rapid occlusion at birth. (**a**) Differential expression of ADAMTS proteases and large CS-proteoglycans during development results in a bilayered artery with a hydrated proteoglycan-rich inner layer and most contractile SMCs located in the outer layer, contrasting with the umbilical vein (see key at top of figure identifying the illustrated major elements). (**b**) At birth, SMC contraction in the outer layer and fluid movement-induced inner layer buckling redirects the inner layer into the lumen. The single-layered vein does not undergo buckling. (**c**) Umbilical artery occlusion at birth prevents neonatal exsanguination, whereas the patent vein allows a final transfusion from the placenta.

## Biomechanical and computational analysis

The umbilical artery and vein were obtained at E18.5 from mouse embryos (n = 4) following approval by the Yale University IACUC (protocol no 2018–11508), then mounted within a custom computer-controlled biaxial device designed specifically for biomechanical testing of murine vessels (*Gleason et al., 2004*). Vessel maintenance, pre-conditioning, biaxial loading protocols and data collection are described in Appendix 1. The umbilical artery was modeled computationally as a thick-walled, bilayered cylindrical tube subjected to swelling of the GAG-rich inner layer and active

contraction of the smooth muscle-rich outer layer; the model also included a passive contribution of extracellular matrix as revealed by biomechanical tests.

## Statistical analysis

Statistical analyses were carried out using GraphPad Prism analytical software (versions 6–8, Graph-Pad, San Diego, CA) in determining statistical significance using two-tailed Student's $t$-test. Statistical details including $N$ and p values are provided in each corresponding figure legend. Statistical analyses for microarray gene expression were performed using Affymetrix's Transcriptome Analysis Console (TAC 4.0) through the RMA-SST sketch algorithm and R version 3.5.2. Fold changes were calculated by an empirical Bayes ANOVA method through the TAC 4.0 software. Details of these and additional methods are provided in Appendix 1.

## Acknowledgements

We acknowledge the Paul Scherrer Institut, Villigen, Switzerland for provision of synchrotron radiation beamtime at the TOMCAT beamline X02DA of the SLS and thank Goran Lovric for assistance. We are grateful to the Disease Investigations team at San Diego Zoo Global for use of sections from the Benirschke archive, and to the late Dr. Kurt Benirschke for collecting the animal umbilical cords used in this study.

## Additional information

### Funding

| Funder | Grant reference number | Author |
|---|---|---|
| National Institutes of Health | HL107147 | Suneel S Apte |
| National Institutes of Health | HL141130 | Suneel S Apte |
| American Heart Association | 17DIA33820024 | Suneel S Apte |
| Sabrina's Foundation | | Elliot H Philipson |
| National Children's Study | Formative Research Project L01-3-RT-01-E | Martina Veigl David Sedwick |
| Mark Lauer Pediatric Research Grant | | Sumeda Nandadasa |
| National Institutes of Health | CA43703 | Martina Veigl |
| Swedish Heart-Lung Foundation | | Karin Tran-Lundmark |
| National Children's Study | Contract # HHSN272500800009C | Martina Veigl David Sedwick |

The funders had no role in study design, data collection and interpretation, or the decision to submit the work for publication.

### Author contributions

Sumeda Nandadasa, Conceptualization, Data curation, Formal analysis, Validation, Investigation, Visualization, Methodology, Writing - original draft, Writing - review and editing; Jason M Szafron, Conceptualization, Formal analysis, Investigation, Visualization, Methodology, Writing - review and editing; Vai Pathak, Data curation, Formal analysis, Visualization, Writing - review and editing; Sae-Il Murtada, Formal analysis, Visualization, Methodology, Writing - review and editing; Caroline M Kraft, Anna O'Donnell, Christian Norvik, Karin Tran-Lundmark, Investigation, Writing - review and editing; Clare Hughes, Nancy B Schwartz, Resources, Writing - review and editing; Bruce Caterson, Resources; Miriam S Domowicz, David Sedwick, Resources, Investigation, Writing - review and editing; Martina Veigl, Resources, Methodology, Writing - review and editing; Elliot H Philipson, Conceptualization, Resources, Supervision, Funding acquisition, Investigation, Writing - original draft, Writing - review and editing; Jay D Humphrey, Conceptualization, Data curation, Formal

analysis, Supervision, Funding acquisition, Visualization, Writing - original draft, Writing - review and editing; Suneel S Apte, Conceptualization, Data curation, Supervision, Funding acquisition, Investigation, Methodology, Writing - original draft, Project administration, Writing - review and editing

## Author ORCIDs
Sumeda Nandadasa  https://orcid.org/0000-0002-4954-6376
Jason M Szafron  https://orcid.org/0000-0001-9476-5175
Clare Hughes  http://orcid.org/0000-0003-4726-5877
Miriam S Domowicz  http://orcid.org/0000-0001-7860-4427
Jay D Humphrey  https://orcid.org/0000-0003-1011-2025
Suneel S Apte  https://orcid.org/0000-0001-8441-1226

## Ethics

Human subjects: Human umbilical cord samples were collected under an IRB exemption (EX-0118) from Cleveland Clinic for use of discarded tissue without patient identifiers. These cords were used for histological/immunohistologic analysis, in situ hybridization, and transcriptomics of inner vs outer umbilical artery TM. For microarray analysis of umbilical cord artery versus vein, human umbilical cords were collected separately through the National Children's Study under University Hospitals-Case Medical Center approved IRB protocol 01-11-28.

Animal experimentation: This study was performed in strict accordance with the recommendations in the Guide for the Care and Use of Laboratory Animals of the National Institutes of Health. All of the animals were handled according to approved institutional animal care and use committee (IACUC) protocols: 18-1996 and 18-2045 (Cleveland Clinic IACUC), 2018-11508 (Yale University IACUC) and 43751 (University of Chicago IACUC).

## Decision letter and Author response
Decision letter https://doi.org/10.7554/eLife.60683.sa1
Author response https://doi.org/10.7554/eLife.60683.sa2

# Additional files

## Supplementary files
• Supplementary file 1. Microarray comparison of transcriptome of the human umbilical artery and vein.

• Supplementary file 2. Microarray comparison of the transcriptome of the inner umbilical artery tunica media with the outer umbilical artery tunica media.

• Transparent reporting form

## Data availability
All data generated or analysed during this study are included in the manuscript and supporting files.

The following datasets were generated:

| Author(s) | Year | Dataset title | Dataset URL | Database and Identifier |
| --- | --- | --- | --- | --- |
| Nandadasa S, Szafron JM, Pathak V, Murtada S-I, Kraft CM, O'Donnell A, Norvik C, Hughes C, Caterson B, Domowicz MS, Schwartz NB, Tran-Lundmark K, Veigl M, Sedwick D, Philipson EH, Humphrey JD, Apte SS | 2020 | Human umbilical cord artery inner tunica media vs outer tunica media. | http://dx.doi.org/10.5061/dryad.4j0zpc88k | Dryad Digital Repository, 10.5061/dryad.4j0zpc88k |

| Nandadasa S, Szafron JM, Pathak V, Murtada S-I, Kraft CM, O'Donnell A, Norvik C, Hughes C, Caterson B, Domowicz MS, Schwartz NB, Tran-Lundmark K, Veigl M, Sedwick D, Philipson EH, Humphrey JD, Apte SS | 2020 | Human umbilical cord artery vs vein | http://dx.doi.org/10.5061/dryad.hdr7sqvfs | Dryad Digital Repository, 10.5061/dryad.hdr7sqvfs |

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

## Appendix 1

### Detailed methods

#### Human tissue

After delivery of the neonate, the umbilical cord was doubly clamped and cut. The time from delivery to cord clamping varied between deliveries and in general ranged from 5 to 30 s. Two additional clamps were placed on the cord after a fetal blood sample was obtained for Rhesus typing, and scissors were used to cut between these two clamps to obtain a segment of umbilical cord for analysis. Umbilical cord samples were fixed in 4% paraformaldehyde for 48 hr at 4°C and embedded in paraffin. All human umbilical cord sections analyzed contained all three vessels and both arteries were imaged in addition to the vein. For microarray analysis of umbilical artery versus vein, human umbilical cords were separately collected as a part of the National Children's Study (NCS) (UHCMC IRB# 01-11-28). Immediately upon delivery, cords were sectioned into 1-inch segments (for a total of 3 segments per cord at 0 hr) and flash frozen in liquid nitrogen. *RNA Later-ICE* (Ambion) was used to maintain nucleic acid viability during freeze-thaw of tissue. The umbilical vein and both umbilical arteries were dissected from each segment for homogenization. RNA from homogenized vein and artery tissue was extracted using Qiagen's RNeasy nucleic acid isolation kit. For microarray analysis of the inner versus outer arterial tunica media, human umbilical arteries were dissected immediately upon neonatal delivery and sectioned into 1-inch segments (for a total of three segments per artery) and their inner and outer tunica media were carefully dissected under a dissecting microscope and flash frozen in liquid nitrogen in Trizol reagent (Ambion). RNA from homogenized arteries was extracted using the chloroform-isopropanol precipitation method.

#### Histological and immuno-staining

Seven-micron-thick paraffin sections were collected using a Leica RM 2255 microtome and stained with hematoxylin and eosin, Alcian blue, or Masson's trichrome after deparaffinization. For immunostaining, sections were deparaffinized, and boiled in citrate buffer (10 mM citric acid, 0.05% Tween 20, pH 6.0) for 90 s for antigen retrieval, washed with PBST and blocked in 10% normal goat serum before incubation with the following primary antibodies overnight at 4°C: Cy3-conjugated smooth muscle α-actin (α-SMA) (1:400, Sigma C6198), anti-smooth muscle myosin heavy chain (SMMHC) (1:400 Kamiya Biomedical, MC352), anti-serine[20]-phosphorylated myosin light chain (pMLC) (Abcam, ab2480 1:200), anti-phospho histone H3 (ser10)(1:200, Millipore, 06–570), anti-chondroitin sulfate (7D4) antibody (*Sorrell et al., 1990*) (1:200), FITC-conjugated anti-heparan sulfate (10E4) antibody (1:200, US biological H-1890), polyclonal anti-versican (*Foulcer et al., 2014*) (anti-VC; 1:400), polyclonal anti-versican GAG-β (1:400, Milipore Sigma, AB1033), anti- DPEAAE (versican V0/V1 neo epitope antibody, 1:200, Invitrogen, PA1-1748A), polyclonal anti-aggrecan (1:400, Milipore Sigma, AB1031), anti- NITEGE (aggrecan neo epitope antibody, 1:200, Invitrogen, PA1-1746), rat monoclonal anti-endomucin (1:400, Invitrogen, 14-5851-85), anti-Sox9 (1:200, Milipore Sigma Ab5535). Alexa-488 or Alexa-568 conjugated secondary antibodies against rabbit and mouse IgG, respectively, were used at 1:400 dilution. Vectashield mounting medium (H-1200) contained DAPI for staining nuclei. All images were taken using an Olympus BX51 microscope connected to a Leica DFC 7000T camera using bright field or fluorescence modes. Multi-channel fluorescent images were merged using the NIH Image J software.

#### Synchrotron-based phase contrast micro-CT

Imaging of arteries and veins from three umbilical cords was performed at the X02DA TOMCAT beamline of the Swiss Light Source at the Paul Scherrer Institute (Villigen, Switzerland). A 4x magnifying objective was used, resulting in a field-of-view of $4.2 \times 3.5$ mm$^2$ and an effective pixel size of $1.63 \times 1.63$ μm$^2$. For each scanned vessel, a stack of 1080 tomographic images was acquired. Data analysis was performed using NIH Image J and Amira. Amira allowed for visualization of the vessels from any angle and images were created by combining two different tomographic imaging planes at a 90-degree angle. NIH Image J was used for creating *Figure 1—videos 1* and *2*.

## Microarray analysis of human umbilical cords

NCS samples to evaluate changes in expression in artery versus vein were run on the Affymetrix Hu-Gene U219 microarray Peg-plate; 150 ng of input RNA from each sample was labeled using an Affymetrix 3' IVT labeling protocol. Samples hybridized to the PEG arrays were washed, stained, and scanned by the Affymetrix Gene Titan. RNA from two of the four cords was also evaluated on the Affymetrix Hu-Gene 1.1 ST Peg-plate microarray, using the whole transcriptome (WT) labeling protocol also starting with 150 ng of input total RNA. The hybridized PEG Arrays were washed, stained, and scanned by the Affymetrix Gene Titan according to standard protocols. RNA (150 ng) from inner and outer TM samples was labeled using Affymetrix's WT PLUS protocol. Labeled samples were hybridized overnight to Affymetrix Hu-Gene 2.0 ST microarray cartridges. The sample was removed, then the microarray cartridges were washed and stained on the Affymetrix GeneChip Fluidics Station450 and scanned by the Affymetrix GeneChip Scanner 3000. Total RNA was labeled using Affymetrix's FLASH Tag Protocol. Labeled samples were hybridized overnight to Affymetrix GeneChip miRNA 2.0 Array, which interrogates all the mature human miRNA sequences in miRBase Release 20microRNAs. The sample was then removed and the microarray cartridges were washed and stained on the Affymetrix GeneChip Fluidics Station450 and scanned by the Affymetrix GeneChip Scanner 3000. Gene expression analysis for each microarray study was performed using Affymetrix's Transcriptome Analysis Console (TAC 4.0) through the RMA-SST sketch algorithm and R version 3.5.2. Fold changes were calculated by an empirical Bayes ANOVA method through the TAC 4.0 software. Parameters for gene expression changes include a p-value$\leq$0.05 and a fold change value $\geq$1.5 and$\leq-$1.5. Data collected from samples labeled by different labeling protocols, obtained using different scanners or hybridized to different arrays were analyzed as independent data sets.

## Additional details of transgenic mice

The *Adamts1* transgenic allele used here (referred to as *Adamts1*$^-$) was generated by Deltagen Inc (San Carlos, CA; Deltagen identifier T1288; MGI:5427602 (B6;129P2-Adamts1 < tm1Dgen>/H;)) by inserting an IRES-lacZ-neomycin resistance gene cassette into intron 1. RT-PCR using PCR primers bridging exon 1 and exon two showed that the insertion eliminated gene expression (Deltagen, unpublished data). The allele was deposited in the MRC Harwell, Frozen Embryo and Sperm Archive (Harwell, UK). Frozen *Adamts1*$^{+/-}$ embryos harvested at the two-cell stage of development from matings of wild-type C57BL/6 females with hemizygous males at MRC Harwell were obtained under an academic use license signed by the Cleveland Clinic with Deltagen. Frozen embryos were implanted into pseudo-pregnant female recipient mice at the Case Transgenic and Targeting Facility (Cleveland, OH). Subsequently, the *Adamts1*$^-$ allele was crossed into the C57BL/6 strain for at least 10 generations and is maintained in this strain. The *Acan*$^{cmd-Bc}$ allele (referred to here as *Acan*$^{+/-}$) was back-crossed to C57BL/6 for over 20 generations at the University of Chicago (*Lauing et al., 2014*) and subsequently transferred to the Cleveland Clinic Lerner Research Institute. β-Galactosidase staining of *Adamts1*$^{+/-}$ umbilical cords was done as previously described (*McCulloch et al., 2009*). Both β-galactosidase staining and genotyping were used to identify *Adamts1*$^{+/-}$ mice obtained from hemizygous with wild-type matings, whereas PCR genotyping was used to distinguish the three genotypes possibly arising from crosses of *Adamts1* or *Acan* hemizygous animals which were used to generate homozygous embryos.

For genotyping, genomic DNA was isolated from clipped toes between 7–10 days after birth or from embryo tails using 100 μl or 50 μl Direct PCR Tail Lysis reagent, respectively (Viagen, catalog number 102 T) containing 1 μl of proteinase K (Milipore Sigma, catalog number 3115879001) followed by digestion overnight at 55˚C. Alternatively, tail DNA was extracted using QuickExtract DNA (Lucigen) at 65˚C for 15 min and reaction was stopped by incubation at 100˚C for 5 min.

The targeted *Adamts1* allele was detected by PCR using forward primer 5' GGGCCAGCTCATTCCTCCCACTCAT 3' and reverse primer 5'GCCATCGGGGTCAGCTTTTCAAATG 3' (generating a 356 bp product) and the wild-type *Adamts1* allele was detected using forward primer 5' GGTTGTAGTTTCGCGCTGAGTTTTG3' and reverse primer 5'GCCATCGGGGTCAGCTTTTCAAATG 3' (generating a 189 bp product). *Acan* genotyping was performed using the following primer pairs: *Acan*$^{cmd}$ forward primer 5' ATCAAGACCCTCAGCTTTTATTAATCTTTA 3' and reverse primer 5' CATAAGATGAGAGGAGATGGTTTAGAGTAT 3' (expected product 811 bp); *Acan* wild type allele

forward primer 5′ TCCTATTTACACAAAGTCTGAAATTAATGC 3′ and reverse primer 5′ GAGAA TTGGCTATAGCTGTTTATGACTC 3′ (expected product 332 bp).

## RNA in situ hybridization

Six-micron-thick paraffin sections were probed with RNAscope probes according to manufacturer's guidelines. The following probes were used: *ACAN* (Advanced Cell Diagnostics (ACD) Cat. No. 506841), *VCAN-exon 8* (ACD Cat. No. 452241), *ADAMTS1* (ACD Cat. No. 524501), *ADAMTS4* (ACD Cat. No. 537341), *ADAMTS5* (ACD Cat. No. 427611), *ADAMTS9* (ACD Cat. No. 445321), *Acan* (ACD Cat. No. 439101), *Vcan-exon 8* (ACD Cat. No. 428321), *Adamts1* (ACD Cat. No. 463361), *Adamts4* (ACD Cat. No. 497161), *Adamts5* (ACD Cat. No. 427621) and *Adamts9* (ACD Cat. No. 400441). The probes were detected using RNAscope 2.5 HD Red reagent kits (ACD Cat. No. 322350) essentially as recently described (*Mead and Apte, 2020*).

## Biomechanical characterization of mouse umbilical arteries and veins

Umbilical cords were excised and separated from the placenta from wild-type C57BL/6J mice at embryonic day E18.5 (n = 4). The umbilical arteries and veins were identified by opening the abdomen of the embryo and locating an artery by its connection to the iliac artery and a vein by its connection to the inferior vena cava through the ductus venosus. After separating and cleaning the umbilical arteries and veins from excessive adipose tissue, the intact vessels were cannulated on custom-drawn glass micropipettes, secured with silk sutures at each end, and mounted within a custom computer-controlled biaxial test device designed specifically for testing murine vessels. These cylindrical specimens were then immersed in a Krebs-Ringer bicarbonate solution (Krebs) and oxygenated with 95% $O_2$ and 5% $CO_2$ while maintained at 37°C. The lumen of the umbilical artery closed immediately after separating the umbilical cord from the placenta but was relaxed by acclimation within the testing chamber for 5–15 min at 10 mmHg and wash-out with Krebs solution up to three times. The vessels were then subjected to two isobaric (luminal pressure of 5 mmHg, then 10 mmHg) - axially isometric (fixed specimen-specific in vivo axial stretch) contractions by adding 100 mM KCl to the bath to ensure viability of the SMC. The transmural organization of the vessel wall was monitored during different stages of contraction using an optical coherence tomography (OCT) system having an axial (depth) resolution <7 microns and lateral resolution of 8 microns (Callisto Model, Thorlabs, Newton, NJ).

For the subsequent passive tests, the normal Krebs solution was replaced with a $Ca^{2+}$-free Krebs solution. The vessels were preconditioned via four cycles of pressurization (artery: 0–40 mmHg and vein: 0–25 mmHg) while held fixed at their individual in vivo axial stretch. Subsequently, the vessels were subjected to a series of seven biaxial testing protocols: cyclic pressurization over ranges noted above at three different fixed values of axial stretch and cyclic axial stretching at four different fixed values of luminal pressure (artery at 10, 20, 30 or 40 mmHg and vein at 1, 2, 10 or 20 mmHg). Data collected online included outer diameter, luminal pressure, axial length, and axial force, which facilitated robust parameter estimation of the biaxial biomechanical behavior (*Ferruzzi et al., 2015*). Toward this end, we used a validated 'four-fiber family' type constitutive relation (see below) that has proved useful in characterizing murine arteries (systemic and pulmonary) and veins.

Note that these biaxial data were used to build a baseline, bilayered cylindrical model for the purposes of parametrically exploring possible effects of different levels of smooth muscle contractility in the outer tunica media, GAG-induced swelling of the inner tunica media, and possible buckling of this innermost layer in silico. Hence, rather than focus on specimen-to-specimen differences, we sought 'mean' properties of the normal murine umbilical artery. Data from all seven passive biaxial testing protocols from all four mice were grouped as a single data set and best-fit values of the associated eight material parameters were determined simultaneously by minimizing the sum-of-the-squares differences between predicted and measured pressures and axial forces, as described previously (*Ferruzzi et al., 2015*). To facilitate a global, rather than local, minimization, we used multiple randomly generated initial guesses for the parameter estimation, accomplished using the MATLAB routine (lsqnonlin function). It is important to note that best-fit parameters in exponential constitutive relations need not be unique, hence what is most important is that together they yield correct values of biaxial stress (verified by comparison to measured values) and predict appropriate levels of

material stiffness and stored energy. Finally, it is noted that initial parameter estimation was based on data from a representative specimen rather than all data combined; the subsequent model simulations were similar in both cases, suggesting that the baseline model captured salient features as desired.

## Computational modeling of the umbilical artery

### Swelling and contraction

The traction-free, non-swollen configuration $(R, \Theta, Z)$ is treated as the reference with inner radius, interfacial radius, and outer radius denoted $[A, B, C]$, respectively. Swelling is accounted for via previously outlined methods (*Szafron et al., 2017*; *Demirkoparan and Pence, 2007*), which leads to a residually stressed, traction-free configuration with corresponding material points mapped to $[A^*, B^*, C^*]$ in $(R^*, \Theta^*, Z^*)$. The final loaded configuration, pressurized and axially stretched, is then characterized by $[a, b, c]$ in $(r, \theta, z)$.

Consider a deformation from $(R, \Theta, Z)$ to $(R^*, \Theta^*, Z^*)$ via the deformation gradient tensor $\mathbf{F}^* = \mathrm{diag}(\partial R^*/\partial R, R^*/R, \Lambda_z^*)$, where volume change is imposed by $\det \mathbf{F}^* = \nu^* = v/V$ with $\nu^*$ the ratio of volume $v$ in $(R^*, \Theta^*, Z^*)$ to volume $V$ in $(R, \Theta, Z)$. With $\nu^* = 1$, there is no swelling and the vessel is considered to have no residual stresses; in contrast, $\nu^* > 1$ (expansion) and $\nu^* < 1$ (shrinkage) yields self-equilibrating wall stresses in the absence of external loading. Note that $\partial R^*/\partial R = R\nu^*/\Lambda_z^* R^*$. Further deformation to any loaded configuration $(r, \theta, z)$ from the swollen configuration $(R^*, \Theta^*, Z^*)$ is then described by $\mathbf{F}^P = \mathrm{diag}(\partial r/\partial R^*, r/R^*, \Lambda_z)$, where the assumption of incompressibility during transient external loading requires $\det \mathbf{F}^P = 1$, thus $\partial r/\partial R^* = R^*/(\Lambda_z r)$. A multiplicative decomposition of the deformations yields $\mathbf{F} = \mathbf{F}^P \mathbf{F}^* = \mathrm{diag}(R\nu^*/\lambda_z r, r/R, \lambda_z)$ with $\lambda_z = \Lambda_z^* \Lambda_z$ for convenience. The matrix component $F_{11}$ can also be expressed in terms of the original derivatives, such that $(\partial r/\partial R^*)(\partial R^*/\partial R) = R\lambda_z \nu^*/r$, which allows us to apply the chain rule and integrate $\int_a^r r \partial r = \int_A^R \nu^* R/\lambda_z \partial R$ to find any radial point $r$ within the vessel wall.

The vessel is assumed to be quasi-equilibrated in any state, such that linear momentum balance requires $\mathrm{div}\, \mathbf{t} = 0$, where $\mathbf{t}$ is the Cauchy stress tensor. Circumferential and axial equilibrium is satisfied identically at each $(r, \theta, z)$. Radial equilibrium requires $\partial t_{rr}/\partial R + (t_{rr} - t_{\theta\theta})/r = 0$. The Cauchy stress is specialized as $\mathbf{t} = \mathbf{t}^{ex} - p\mathbf{I}$ with $p$ the Lagrange multiplier enforcing incompressibility during transient loading, $\mathbf{I}$ the identity tensor, and $\mathbf{t}^{ex}$ the 'extra' part of the stress due to deformation and the constitutive response. Integration yields

$$P = \int_a^b \left(t_{\theta\theta}^{ex} - t_{rr}^{ex}\right)/r dr + \int_b^c \left(t_{\theta\theta}^{ex} - t_{rr}^{ex}\right)/r dr$$

where $P$ is the transmural pressure across the vessel wall, with $P > 0$ indicating internal pressurization. We also calculate the overall axial load required for overall equilibrium,

$$L = \pi \int_a^b \left(2t_{zz}^{ex} - t_{\theta\theta}^{ex} - t_{rr}^{ex}\right) r dr + \pi \int_b^c \left(2t_{zz}^{ex} - t_{\theta\theta}^{ex} - t_{rr}^{ex}\right) r dr + P\pi a^2$$

See *Figure 7d*. The equilibrium problem is solved iteratively for the loaded inner radius $a$ for each luminal pressure $P$ and axial extension $\lambda_z$.

Constitutively, the extra part of the Cauchy stress can be computed from a stored energy density function $W$ for the vessel, with $\mathbf{t} = 2\mathbf{F}(\partial W/\partial \mathbf{C})\mathbf{F}^T/\det \mathbf{F} - p\mathbf{I}$ and $\mathbf{C} = \mathbf{F}^T\mathbf{F}$ the right Cauchy-Green tensor. Due to the microstructure of the umbilical vessels, the GAG-rich inner layer is modeled as a neo-Hookean matrix that can swell (*Demirkoparan and Pence, 2007*), with

$$W = \frac{\mu_1}{2}\left(\mathrm{tr}(\mathbf{C}) - 3\nu^{*\frac{2}{3}}\right) \quad \forall\, r < b,$$

with $\mu_1$ a shear modulus for this inner layer. As there is no evidence of collagen with a preferred orientation or smooth muscle cells capable of contraction within the inner layer, it is considered isotropic and passive. Fewer GAGs are present in the outer layer, but we include the possibility of a swellable matrix for illustrative purposes and to provide radial stiffness. The outer layer is then modeled using a modified four-fiber family model for a passive nonlinear stress-stretch behavior and a Rachev-type model for SMC contractility (i.e. active behavior), with a potential function

$$W = \frac{\mu_2}{2}\left(\text{tr}(\mathbf{C}) - 3\nu^{*\frac{2}{3}}\right) + \sum_{\alpha=1}^{4} \frac{c_1^\alpha}{4c_2^\alpha}\left(\exp\left(c_2^\alpha\left(\lambda^{\alpha^2} - 1\right)^2\right) - 1\right)$$
$$+ \text{T}_{act}\left(\lambda_\theta + \frac{1}{3}\frac{(\lambda_m - \lambda_\theta)^3}{(\lambda_m - \lambda_0)^2}\right) \forall\, r > b$$

where $\mu_2$ is shear modulus for the outer layer, $c_1^\alpha$ and $c_2^\alpha$ are material parameters for each fiber family $\alpha = 1, 2, 3, 4$, $\lambda^\alpha = \text{sqrt}(\mathbf{C}{:}\mathbf{M}^\alpha \otimes \mathbf{M}^\alpha)$, $\text{T}_{act}$ is the magnitude of the active stress, $\lambda_m$ is the stretch at which contraction is maximum, and $\lambda_0$ is the stretch at which contraction ceases (**Baek et al., 2007**). Each fiber family has an orientation vector $\mathbf{M}^\alpha = \sin(\eta^\alpha)\mathbf{e}_\Theta + \cos(\eta^\alpha)\mathbf{e}_Z$ with $\eta^\alpha$ the fiber angle relative to the axial direction. The fiber families are assumed to lie in the circumferential direction ($\alpha = 1$, $\eta^1 = 90°$), the axial direction ($\alpha = 2$, $\eta^2 = 0°$), and symmetric diagonal directions about the $z$-axis with $\eta^\alpha$ fit from the experimental data ($\alpha = 3$, $\eta^3 = \eta^d$ and $\alpha = 4$, $\eta^4 = -\eta^d$). Parameter values are given in **Table 1** based on the aforementioned nonlinear least squares approach (**Ferruzzi et al., 2013**).

## Bifurcation analysis – basic approach

To examine the potential for unstable equilibria (i.e. bifurcations in the solutions) leading to buckling of the wall, consider an incremental deformation added to the finite deformation (**Ogden, 1984**) with a notation similar to that previously used to describe bifurcation behaviors in growing elastic solids (**Li et al., 2011**; **Moulton and Goriely, 2011**), which is mathematically similar to swelling. Quantities related to the intermediate finite deformation are given as $(\cdot)^{(0)}$ while those related to the incremental motions are denoted as $(\cdot)^{(1)}$. The current position $x$ from position $x^{(0)}$ is specified as $x = x^{(0)} + \epsilon u^{(1)}$ with $\epsilon \ll 1$ scaling the displacement $u^{(1)}$, which allows the deformation gradient to be written as $\mathbf{F} = \mathbf{F}^{(0)} + \epsilon\mathbf{H}^{(1)}\mathbf{F}^{(0)}$ with $\mathbf{F}^{(0)} = \mathbf{F}^P\mathbf{F}^*$ the deformation gradient from the reference to the finitely deformed configuration and $\mathbf{H}^{(1)}$ the incremental displacement gradient with respect to that configuration. The nominal stress $\mathbf{P}$, defined through $\mathbf{t} = \mathbf{F}\mathbf{P}/\det\mathbf{F}$, follows as $\mathbf{P} = \mathbf{P}^{(0)} + \epsilon\mathbf{P}^{(1)}$. As noted previously (**Haughton and Ogden, 1978**; **Sanft et al., 2019**), it is convenient to update the reference configuration to the current configuration yielding $\mathbf{P}_0 = \mathbf{P}_0^{(0)} + \epsilon\mathbf{P}_0^{(1)}$, which, along with considering the Lagrange multiplier as $p = p^{(0)} + \epsilon p^{(1)}$, allows us to write $\mathbf{P}_0^{(1)} = \mathfrak{B}{:}\mathbf{H}^{(1)} + p^{(0)}\mathbf{H}^{(1)} - p^{(1)}\mathbf{I}$ with $\mathfrak{B} = \mathbf{F}^{(0)}\left(\partial^2 W/\partial\mathbf{F}^{(0)}\partial\mathbf{F}^{(0)}\right)\mathbf{F}^{(0)}$ the fourth-order stiffness tensor calculated in the finitely deformed configuration and $p^{(1)}$ the increment in the Lagrange multiplier. Linear momentum balance then requires $\text{div}\mathbf{P}_0 = 0$, which is satisfied by the finite deformation, leaving $\text{div}\mathbf{P}_0^{(1)} = 0$ to be resolved. We proceed by assuming a form for the incremental displacement describing buckling as $u^{(1)} = [\text{u}(r, \theta), \text{v}(r, \theta), 0]$, with no incremental motion in the axial direction. Displacements in the $r$ and $\theta$ directions are not a function of $z$. The incremental displacement gradient then becomes

$$\mathbf{H}^{(1)} = \begin{bmatrix} \text{u}_r & \frac{\text{u}_\theta - \text{v}}{r} & 0 \\ \text{v}_r & \frac{\text{u} + \text{v}_\theta}{r} & 0 \\ 0 & 0 & 0 \end{bmatrix},$$

where $(\cdot)_r \equiv \partial(\cdot)/\partial r$ and $(\cdot)_\theta \equiv \partial(\cdot)/\partial\theta$. The incremental deformation is assumed to be isochoric, requiring $\text{tr}\left(\mathbf{H}^{(1)}\right) = 0$. As there is no axial dependence in the incremental deformation, the equilibrium equations reduce to a system of two differential equations in u, v, and $p^{(1)}$. These functions are assumed to have sinusoidal forms in the buckled configuration, where $\text{u} = f(r)\cos(n\theta)$, $\text{v} = g(r)\sin(n\theta)$, and $p^{(1)} = h(r)\cos(n\theta)$ with $n$ the buckling mode (i.e. the number of folds in the inner portion of the vessel wall). Using the incompressibility condition, it is possible to rewrite $g(r)$ in terms of $f(r)$, and the two equilibrium equations can be combined to eliminate $h(r)$, yielding a single fourth order, ordinary differential equation in $f(r)$ (**Haughton and Ogden, 1978**; **Sanft et al., 2019**) of the form

$$A_4 f'''' + A_3 f''' + A_2 f'' + A_1 f' + A_0 f = 0,$$

where coefficients $A_0$-$A_4$ are given below and $(\cdot)' = d(\cdot)/dr$. We assume that the incremental tractions on the inner surface are zero with $\mathbf{n}\cdot\mathbf{t}^{(1)} = 0$, which yields two equations

$$\left[B_{1,3}f''' + B_{1,2}f'' + B_{1,1}f' + B_{1,0}f = 0\right]_{r=a} \text{ and } \left[B_{2,2}f'' + B_{2,1}f' + B_{2,0}f = 0\right]_{r=a},$$

with coefficients $B_{n,m}$ given below. As there is little evidence of buckling in the outer layer, and the Rachev-type contractility model generally yields tensile stresses that would inhibit buckling, we specify that the incremental deformation vanishes at the interface of the two layers with $f(b) = 0$ and $f'(b) = 0$ and that the incremental shear traction is zero (*Yang et al., 2007*), which gives

$$\left[C_{1,2}f'' + C_{1,1}f' = 0\right]_{r=b} \text{ and } \left[C_{2,2}f'' + C_{2,0}f = 0\right]_{r=b},$$

with coefficients $C_{n,m}$ given below.

We used the compound matrix method (*Haughton and Orr, 1997*; *Lindsay and Rooney, 1992*), a modification of the determinantal method commonly used for linear bifurcation analysis (*Haughton and Ogden, 1979*), to solve the differential equation numerically. The fourth order equation above was rewritten as a system of four, first order differential equations, $y' = Ay$, where $y = \left[f, f', f'', f'''\right]^T$ and $A$ is the corresponding coefficient matrix. Boundary conditions at the inner surface and interface were similarly re-written as $[By = 0]_a$ and $[Cy = 0]_b$, respectively. We define two linearly independent initial conditions at $r = a$, which can be integrated to $r = b$ to create linearly independent solutions $y^{(1)}$ and $y^{(2)}$ such that $y = k_1 y^{(1)} + k_2 y^{(2)} = Mk$. For the determinantal method, one iterates on $\mathrm{T}_{act}$ until $\det\left([CM]_b\right) = 0$. However, this method can fail for stiff systems, thus we use Laplace expansions to write a new bifurcation condition equivalent to the original with $\det(CM) = \sum_k |C_k|\phi_k$

where

$$\phi_1 = \begin{vmatrix} y_1^{(1)} & y_1^{(2)} \\ y_2^{(1)} & y_2^{(2)} \end{vmatrix} = (1,2), \quad \phi_2 = \begin{vmatrix} y_1^{(1)} & y_1^{(2)} \\ y_3^{(1)} & y_3^{(2)} \end{vmatrix} = (1,3),$$

and similarly, $\phi_3 = (1,4)$, $\phi_4 = (2,3)$, $\phi_5 = (2,4)$, $\phi_6 = (3,4)$. We evaluate $\mathfrak{C}_k = |C_k|$ with

$$\mathfrak{C}_1 = \begin{vmatrix} C_{1,1} & C_{2,1} \\ C_{1,2} & C_{2,2} \end{vmatrix} = (1,2), \quad \mathfrak{C}_2 = \begin{vmatrix} C_{1,1} & C_{2,1} \\ C_{1,3} & C_{2,3} \end{vmatrix} = (1,3),$$

and similarly, $\mathfrak{C}_3 = (1,4)$, $\mathfrak{C}_4 = (2,3)$, $\mathfrak{C}_5 = (2,4)$, $\mathfrak{C}_6 = (3,4)$, where

$$(n,m) \equiv \begin{vmatrix} C_{1,n} & C_{2,n} \\ C_{1,m} & C_{2,m} \end{vmatrix}.$$

To evaluate the new bifurcation condition, we create a system of equations $\phi' = \mathfrak{A}\phi$ and identify the components of $\phi'$ as, for example,

$$\phi_1' = \begin{vmatrix} y_1^{(1)'} & y_1^{(2)'} \\ y_2^{(1)} & y_2^{(2)} \end{vmatrix} + \begin{vmatrix} y_1^{(1)} & y_1^{(2)} \\ y_2^{(1)'} & y_2^{(2)'} \end{vmatrix} = \begin{vmatrix} \sum_i^4 A_{1i}y_i^{(1)} & \sum_i^4 A_{1i}y_i^{(2)} \\ y_2^{(1)} & y_2^{(2)} \end{vmatrix} + \begin{vmatrix} y_1^{(1)} & y_1^{(2)} \\ \sum_i^4 A_{2i}y_i^{(1)} & \sum_i^4 A_{2i}y_i^{(2)} \end{vmatrix}.$$

The components of $\mathfrak{A}$ can thus be conveniently defined in terms of the original components of $A$ (*Haughton and Orr, 1997*), as listed below. This new system is then integrated from $a$ to $b$ using a fourth-order Runge-Kutta method, and $\mathrm{T}_{act}$ is varied iteratively until the boundary condition at $b$ is satisfied, namely $\left[\sum_k \mathfrak{C}_k \phi_k\right]_b = 0$. Loading conditions, including the volume change $\nu^*$, luminal pressure $P$, and axial stretch $\lambda_z$, can be varied parametrically to understand their effects on the critical value of $\mathrm{T}_{act}$ needed to induce buckling. Note, one may also fix the value of $\mathrm{T}_{act}$ and identify the critical value of a different loading variable of interest.

## Bifurcation analysis – specific functions

For the fourth order governing differential equation for the incremental displacement:

$$A_4 f'''' + A_3 f''' + A_2 f'' + A_1 f' + A_0 f = 0$$

we have

$$
\begin{aligned}
A_0 &= (n^2-1)\left( r^2 \mathfrak{B}''_{r\theta r\theta} + r \mathfrak{B}'_{r\theta r\theta} + (n^2-1)*\mathfrak{B}_{r\theta r\theta} + n^2(\mathfrak{B}_{\theta\theta\theta\theta} - \mathfrak{B}_{rrrr}) \right) \\
A_1 &= r\left( \left( r^2 \mathfrak{B}''_{r\theta r\theta} + r \mathfrak{B}'_{r\theta r\theta} - \mathfrak{B}_{r\theta r\theta} - n^2(\mathfrak{B}_{\theta\theta\theta\theta} + \mathfrak{B}_{rrrr}) \right) - n^2 r(\mathfrak{B}'_{\theta\theta\theta\theta} + \mathfrak{B}'_{rrrr}) \right) \\
A_2 &= r^2\left( r^2 \mathfrak{B}''_{r\theta r\theta} + 7r \mathfrak{B}'_{r\theta r\theta} + 5\mathfrak{B}_{r\theta r\theta} - n^2(\mathfrak{B}_{\theta\theta\theta\theta} + \mathfrak{B}_{rrrr}) \right) \\
A_3 &= r^3\left( 2r \mathfrak{B}'_{r\theta r\theta} + 6\mathfrak{B}_{r\theta r\theta} \right) \\
A_4 &= r^4 \mathfrak{B}_{r\theta r\theta}
\end{aligned}
$$

Note: These coefficients include only the non-zero components of $\mathfrak{B}$ for the considered stored energy density function.

For the fourth order differential equation

$$\mathbf{y}' = \mathbf{A}\mathbf{y}$$

note that

$$
\mathbf{A} = \begin{bmatrix}
0 & 1 & 0 & 0 \\
0 & 0 & 1 & 0 \\
0 & 0 & 0 & 1 \\
-A_0/A_4 & -A_1/A_4 & -A_2/A_4 & -A_3/A_4
\end{bmatrix}
$$

For boundary conditions on the governing equation at the inner surface:

$$\left[ B_{1,3} f''' + B_{1,2} f'' + B_{1,1} f' + B_{1,0} f = 0 \right]_{r=a}$$

we have,

$$
\begin{aligned}
B_{1,0} &= (n^2-1)\left( r\mathfrak{B}'_{r\theta r\theta} + \mathfrak{B}_{r\theta r\theta} \right) \\
B_{1,1} &= r\left( r\mathfrak{B}'_{r\theta r\theta} - (n^2-1)\mathfrak{B}_{r\theta r\theta} - n^2(\mathfrak{B}_{\theta\theta\theta\theta} + \mathfrak{B}_{rrrr}) \right) \\
B_{1,2} &= r^2\left( r\mathfrak{B}'_{r\theta r\theta} + 4\mathfrak{B}_{r\theta r\theta} \right) \\
B_{1,3} &= r^3 \mathfrak{B}_{r\theta r\theta} \\
B_{2,0} &= \mathfrak{B}_{r\theta r\theta}\,(n^2-1) \\
B_{2,1} &= \mathfrak{B}_{r\theta r\theta}\,r \\
B_{2,2} &= \mathfrak{B}_{r\theta r\theta}\,r^2 \\
B_{2,3} &= 0
\end{aligned}
$$

For boundary conditions on the governing equation at the interface:

$$\left[ C_{1,2} f'' + C_{1,1} f' = 0 \right]_{r=b}$$

we have

$$
\begin{aligned}
C_{1,0} &= 0 \\
C_{1,1} &= \mathfrak{B}_{r\theta r\theta}\,r \\
C_{1,2} &= \mathfrak{B}_{r\theta r\theta}\,r^2 \\
C_{1,3} &= 0 \\
C_{2,0} &= \mathfrak{B}_{r\theta r\theta}\,(n^2-1) \\
C_{2,1} &= 0 \\
C_{2,2} &= \mathfrak{B}_{r\theta r\theta}\,r^2 \\
C_{2,3} &= 0
\end{aligned}
$$

Finally, for the compound matrix method component matrix

$$\mathfrak{A} = \begin{bmatrix} A_{11}+A_{22} & A_{23} & A_{24} & -A_{13} & -A_{14} & 0 \\ A_{32} & A_{11}+A_{33} & A_{34} & A_{12} & 0 & -A_{14} \\ A_{42} & A_{43} & A_{11}+A_{44} & 0 & A_{12} & A_{13} \\ -A_{31} & A_{21} & 0 & A_{22}+A_{33} & A_{34} & -A_{24} \\ -A_{41} & 0 & A_{21} & A_{43} & A_{22}+A_{44} & A_{23} \\ 0 & -A_{41} & A_{31} & -A_{42} & A_{32} & A_{33}+A_{44} \end{bmatrix}$$

with the components of $A$ given above.

