## [Decision Letter]

**Acceptance summary:**

We agree that study of the mature umbilical cord has been neglected - especially lacking is understanding how it prevents fetal blood loss during passage from the birth canal. The ground-breaking analyses described in this manuscript, leading to your unprecedented model of umbilical arterial closure at birth, will provide a wealth of new insight into the biology and biomechanics of one of Placentalia's most important vascular structures.

**Decision letter after peer review:**

Thank you for submitting your article "Vascular dimorphism ensured by regulated proteoglycan dynamics favors rapid umbilical artery closure at birth" for consideration by eLife. Your article has been reviewed by Didier Stainier as the Senior Editor, a Reviewing Editor, and two reviewers. The following individuals involved in review of your submission have agreed to reveal their identity: Jessica Wagenseil (Reviewer #2).

The reviewers have discussed the reviews with one another and the Reviewing Editor has drafted this decision to help you prepare a revised submission.

We would like to draw your attention to changes in our revision policy that we have made in response to COVID-19 (https://elifesciences.org/articles/57162). Specifically, we are asking editors to accept without delay manuscripts, like yours, that they judge can stand as eLife papers without additional data, even if they feel that they would make the manuscript stronger. Thus the revisions requested below only address clarity and presentation.

Summary:

A comprehensive and elegant combination of gene/protein expression, cross-species analyses, genetic mutants, and biomechanical testing/computer analysis was used to study a universal but generally neglected biological event common to all Placentalia: rapid umbilical closure at birth. The phenomenon is critical to fetal survival because, as the purveyor of fetal blood to the chorionic bed of the placenta, the artery must rapidly close to prevent ex-sanguination of the fetus as it exits the birth canal. By contrast, the vein carries blood from the chorionic bed to the fetus and thus, at least in this regard, is less significant.

This is a highly significant and universally relevant study to all Placentalia, providing new insight into the cell/developmental biology and biomechanics of one of this group's most important vascular structures. The study has been beautifully executed and generally well presented. Every finding is novel. And now, for the first time, closure of the umbilical cord at birth has an experimental precedent. The authors conclude that evolution has made sure that a healthy placenta permits a healthy transition from fetal to neo-natal life.

Essential revisions:

1) Computational Model

Variability in the experimental data, challenges in testing and fitting material parameters for these vessels, and how that might affect the conclusions of the computational model should be briefly discussed. The authors already have all the data and are using it for the computational model to draw conclusions, but they left out a lot of details for the modeling and did not address how some of the assumptions and fitted parameters used in the model would affect their conclusions.

2) Logic of Presentation

Because the biomechanical testing in Figure 4 assumed similarities between mouse and human umbilical arteries and veins in advance of the confirmation that was ultimately presented in Figure 5 (mouse studies), could the authors reverse the order of presentation, both in the text and in the figures which follow, i.e., place the whole of their (descriptive) mouse data immediately after the large animal analysis (current Figure 3) and before the functional biomechanical testing and computer modeling (current Figure 4)?

3) All Figures:

For all of the Figures involving synchrotron, histological, immunofluorescence, and RNA-ISH, please state in the corresponding legend how many samples were imaged.

4) Figure 1.

a) Given that the human umbilical cord has two arteries and a vein, did the authors ever separate out the arteries for analysis and, if not, why not? Please address this somewhere in the manuscript.

b) Similarly, in the Supplemental methods, "Veins and arteries were dissected...." - please clarify here whether both arteries were dissected.

5) Figure 1—figure supplement 1C,D.

These panels appear to show that the smooth muscle cells (SMC) of both vessels are similarly organized with alternating circumferential and longitudinal layers, but that the major difference between them is fewer layers of SMC in the vein. It is not clear from the text or legend whether the authors conclude this, too - could they clarify their interpretation in the text?

6) Figure 4F.

What do the authors make of the differences in the number of buckles within and between human umbilical cords - statistically significant? Can they say anything about those differences in the other mammals?

7) Figure 5. The mutant analyses:

a) In the text, legends, and figures, the notation is inconsistent for the knockout mouse models: Adamts1-/-, Adamts1 KO, Acan-/-, Acancmd/cmd, Acan mutants, and Acan KO are all used. Please choose a consistent notation for each model.

b) Please provide information on genotyping mutant litters in the Methods/Supplemental Methods, and their ratios at each embryonic day examined, especially as the authors claim that mutants did not show intrauterine growth retardation or death (see Figure 5—figure supplement 1). If possible, please indicate the background resorption level in these strains, which would be evident by having included the genotype of the resorptions.

c) Could the authors group presentation of aggregan and adamts1 mutants rather than intermingle the data within the text? - it was difficult to follow which result correlated with which mutant.

d) Figure 5G. The PHH3 staining examples are not very good. How are they being normalized to calculate a percentage?

8) Figure 4—figure supplement 1.

a) Need SD or SEM.

b) The number of samples used for mechanical testing needs to be included.

c) Please include circumferential and axial stress-strain curves which are directly related to the fitted material parameters used for the modeling.

d) Figure 4—figure supplement 1B, subsection “Differential SMC contraction in the bilayered umbilical arteries and vein”of the text state that the artery has a smaller lumen, citing this panel, but this panel shows the outer diameter, and not the inner one.

e) Figure 4—figure supplement 1C and subsection “Differential SMC contraction in the bilayered umbilical arteries and vein”, text: Please define "distensibility" and "extensibility" as used in the text. At what pressures or axial stretches are you comparing the distensibility and extensibility? How do the axial stretch values in Figure 4—figure supplement 1C compare to the in vivo axial stretches?

9) Figure 5—figure supplement 1.

Please support the important conclusion, subsection “Aggrecan and Adamts1 are necessary for normal umbilical cord morphogenesis” based on this figure that "Neither mutant showed intrauterine growth retardation, and intrauterine death was infrequent, suggesting adequate cord circulation.", with actual measurements, especially as the acan KO mutant in Figure 5—figure supplement 1B appears smaller at E14.5 than its wildtype counterpart - assuming the same magnifications, which should be indicated.

10) Figure 8.

- What are the blue cells? Something other than SMCs?

11) Table 1.

a) As c1 for the diagonal fibers is 2-3 orders of magnitude below the circ and axial fibers, are the diagonal fibers really necessary in the constitutive model?

b) Also, the c2 values are 1-2 orders of magnitude higher than values that the group has published previously for mouse elastic arteries. Can the authors comment on the suitability of the constitutive model for fitting such nonlinear data and differences between the mechanical behavior of umbilical artery/vein and elastic arteries in the mouse?

c) Are the values in Table 1 averages of individual values from multiple arteries, fit from a combination of data from multiple arteries, or representative values from a single artery?

[Editors' note: further revisions were suggested prior to acceptance, as described below.]

Thank you for re-submitting your article "Vascular dimorphism ensured by regulated proteoglycan dynamics favors rapid umbilical artery closure at birth" for re-consideration by eLife. Your revised manuscript has been re-reviewed Didier Stainier as the Senior Editor, a Reviewing Editor, and two reviewers. The following individuals involved in review of your submission have agreed to reveal their identity: Jessica Wagenseil (Reviewer #2).

The reviewers have discussed the reviews with one another and the Reviewing Editor has drafted this decision to help you prepare a revised submission.

We would like to draw your attention to changes in our revision policy that we have made in response to COVID-19 (https://elifesciences.org/articles/57162). Specifically, we are asking editors to accept without delay manuscripts, like yours, that they judge can stand as eLife papers without additional data, even if they feel that they would make the manuscript stronger. Thus the revisions requested below only address clarity and presentation.

Summary:

In their revised manuscript concerning the mechanism of umbilical arterial closure at birth in Placentalia, Nandadasa et al., have satisfactorily addressed the majority of the reviewers' concerns. However, there remain two major concerns: (1) the mouse mutant analyses, and (2) the number of specimens used per experiment; a small number of minor revisions; and requested changes to the Abstract, in accord with eLife's policies.

Essential revisions:

1) Genetic mutants. The reviewers had requested the following (copied from the previous letter to the authors): (b) Please provide information on genotyping mutant litters in the Materials and methods section/Supplemental methods, and their ratios at each embryonic day examined, especially as the authors claim that mutants did not show intrauterine growth retardation or death (see Figure 5—figure supplement 1). If possible, please indicate the background resorption level in these strains, which would be evident by having included the genotype of the resorptions.

Mouse mutant Adamts-/-. As the authors did not provide their method for genotyping this mutant strain, as requested, more investigation was needed on the part of the reviewers to understand what exactly this strain is to consider why the authors ignored the request.

Background (Oller et al., 2017): The Adamts-/- mutant used in this study was described by Oller et al., 2017 as an insertion of a lacZ-bearing cassette into intron 1 of the gene. According to this previous paper, not only does this insertion reveal where Adamts1 is expressed via staining for ß-galactosidase activity, but in its hemizygous state, the insertion also causes a reduction in both mRNA and protein. In its homozygous state, ß-galactosidase activity is still detectable, but the Adamts1 protein is not (Figure 1a of Oller et al., 2017). Oller et al did not provide details on how the animals were maintained and mated (which they really should have been asked to do, alas), but they did explain how their litters were genotyped, the specific sequence used in those PCR genotyping experiments, and the genotypic ratios of their animals at weaning (Supplemental Figure 1b of Oller et al., 2017).

In the current study, the authors distinguished hemizygotes and homozygotes in the figure panels (e.g., Figure 4C versus the other panels), and wildtype and homozygotes in the others that relate to the mutant analysis, implying that they genotyped this material. However, they ignored the reviewers' procedural request, which is repeated and expanded as follows:

Please provide information on

i) genotyping method used, including the exact DNA sequence for PCR analysis;

ii) genotypic ratios according to gestational day;

iii) the genetic background on which the Adamts animals were maintained (according to Oller et al., it seems to be a B/6 background, but please confirm);

iv) the parental genotypes used to produce the specific genotypes (perhaps hemizygous by wildtype matings produced hemizygous embryos detectable by X-gal staining, whilst homozygous embryos were obtained by crossing hemizygous animals and using PCR genotyping to distinguish the three genotypes? - please confirm/clarify);

v) how gestational age was determined. Were timed matings used? if not timed matings, then how was gestational age determined?;

vi) the protocol used for X-gal staining the hemizygotes.

While Oller et al., 2017 used this transgenic mouse line, they did not create it. Please clarify the specific origin of this mutant mouse strain - that information was impossible to locate on the EMMA site.

It would be most helpful for the reader if the authors would introduce the Adamts mutant in the Results section by summarizing Oller et al.'s results concerning the levels of mRNA versus protein in hemizygotes and homozygotes, as under "Background", above.

Finally, if the authors can, would they comment on whether hemizygotes exhibited foreshortened umbilical cords, too, and did their lengths fall between those of the wildtype and homozygous mutants?

Mouse mutant Acan-/-.

Although the authors provided the genotypic ratios of the Acan-/- mutants as requested by the reviewers (new Figure 4—figure supplement 1 Panel 4C), they did not indicate the genotyping procedure. Please add it to Detailed Methods, to include how the DNA was obtained, and the DNA sequence used to PCR the littermates' DNA.

The mutant allele was originally described by Krueger et al., 1999, but from where did the authors procure this mouse strain?

Litters were obtained at E12.5, E14.5, and E18.5; please indicate how matings were carried out to ascertain the timing of gestation, including parental genotypes that produced the Acan litters.

2) The reviewers had requested the number of specimens (n) for every experiment. The following are still missing:

Figure 1D. n, the number of immunostained specimens?

Figure 2B, line 545. "n=3 umbilical cords" - for each probe?, or for both?

Figure 4—figure supplement 1.

S4b. n = ?

S4d. n = ?

---

## [Author Response]

Summary:A comprehensive and elegant combination of gene/protein expression, cross-species analyses, genetic mutants, and biomechanical testing/computer analysis was used to study a universal but generally neglected biological event common to all Placentalia: rapid umbilical closure at birth. The phenomenon is critical to fetal survival because, as the purveyor of fetal blood to the chorionic bed of the placenta, the artery must rapidly close to prevent ex-sanguination of the fetus as it exits the birth canal. By contrast, the vein carries blood from the chorionic bed to the fetus and thus, at least in this regard, is less significant.This is a highly significant and universally relevant study to all Placentalia, providing new insight into the cell/developmental biology and biomechanics of one of this group's most important vascular structures. The study has been beautifully executed and generally well presented. Every finding is novel. And now, for the first time, closure of the umbilical cord at birth has an experimental precedent. The authors conclude that evolution has made sure that a healthy placenta permits a healthy transition from fetal to neo-natal life.Essential revisions:1) Computational ModelVariability in the experimental data, challenges in testing and fitting material parameters for these vessels, and how that might affect the conclusions of the computational model should be briefly discussed. The authors already have all the data and are using it for the computational model to draw conclusions, but they left out a lot of details for the modeling and did not address how some of the assumptions and fitted parameters used in the model would affect their conclusions.

Thank you for this important comment. Our primary goal was to understand how the umbilical artery closes at birth, which necessitated computational modeling of swelling, contraction, and buckling. We thus focused on the model, though indeed we needed new data to inform the model. We now provide much more detail on these data and how they were collected. We nevertheless emphasize that we sought a general mechanism, not subject-to-subject differences, hence we used “mean data” to inform the modeling, which was then explored via extensive numerical parametric studies in silico.

We used a custom computer-controlled biaxial testing system designed specifically for testing murine arteries, which has proven highly reliable. We used a nonlinear, anisotropic constitutive relation that we have found to be robust in describing diverse murine arteries (systemic and pulmonary) and veins, a relation that has been validated independently by multiple groups. Best-fit material parameters for the 4-fiber family constitutive model (Table 1) were determined using a standard nonlinear regression approach in MATLAB (lsqnonlin function), where differences in predicted vs. measured pressure-diameter data and force-length data were minimized. Multiple random initial guesses were used to ensure convergence of the nonlinear regression to the same minimum. The primary parameters were necessarily obtained from passive, tensile data, yet a key parameter in this study of buckling was the compressive stiffness. We thus studied parametrically the effects of the associated neoHookean parameter in the inner layer, which is greater in compression due to the increased presence of GAGs identified in the immunoassays. Table 1 reports values from our data analysis and many preliminary simulations.

Computational results were initially presented for a representative umbilical artery sample, but, motivated by the reviewer’s excellent suggestion, we went back and fit simultaneously all of the data from all 4 umbilical artery samples, hence yielding best-fit parameters for the truly mean behavior (which is different from using mean values of the individually determined parameters, which we never do). There were some changes in the individual parameter values (based on the mean data) from those used previously (for a single representative sample), particularly for the one fiber-family parameters (which resolved a concern of the reviewer noted below), but these changes in parameter values did not change any of the conclusions from resulting simulation outputs, suggesting further that the buckling phenomenon occurs across different parameter values and is required for closure.

2) Logic of PresentationBecause the biomechanical testing in Figure 4 assumed similarities between mouse and human umbilical arteries and veins in advance of the confirmation that was ultimately presented in Figure 5 (mouse studies), could the authors reverse the order of presentation, both in the text and in the figures which follow, i.e., place the whole of their (descriptive) mouse data immediately after the large animal analysis (current Figure 3) and before the functional biomechanical testing and computer modeling (current Figure 4)?

We agree with the reviewer’s suggestion and have made the requested changes to both the figures and the text. The mouse umbilical cord data is now presented immediately after the large mammal data followed by computational modeling.

3) All Figures:For all of the Figures involving synchrotron, histological, immunofluorescence, and RNA-ISH, please state in the corresponding legend how many samples were imaged.

We have added the missing *n* information to the legend of each figure panel.

4) Figure 1.a) Given that the human umbilical cord has two arteries and a vein, did the authors ever separate out the arteries for analysis and, if not, why not? Please address this somewhere in the manuscript.

In all our experiments the two arteries of human cords appeared identical and indistinguishable in histology and immunostaining. Since it is not possible to identify a distinction between the arteries (assigning for example in each cord, an artery A or B designation), such a comparison would have little basis. Therefore, we did not dissect the two arteries and study them individually. All human umbilical cord sections analyzed contained three vessels and both arteries were imaged in addition to the vein. We have added this information to the methods section of the manuscript. The only differences observed between the two arteries from an individual cord are the number of folds in the areas analyzed. We have illustrated this information in Figure 7F (cord #12,15,16 and 19).

b) Similarly, in the Supplemental Methods, "Veins and arteries were dissected...." - please clarify here whether both arteries were dissected.

We have updated the supplemental methods section to reflect both arteries were used.

5) Figure 1—figure supplement 1C,D.These panels appear to show that the smooth muscle cells (SMC) of both vessels are similarly organized with alternating circumferential and longitudinal layers, but that the major difference between them is fewer layers of SMC in the vein. It is not clear from the text or legend whether the authors conclude this, too - could they clarify their interpretation in the text?

We agree with the reviewer’s interpretation. Histologically, the major difference observed between the arteries and the vein was the thickness of the tunica media, with the vein having fewer SMC layers overall, as clarified in the revised text.

6) Figure 4F.What do the authors make of the differences in the number of buckles within and between human umbilical cords - statistically significant? Can they say anything about those differences in the other mammals?

Our conclusions were limited to the very short length of each vessel analyzed (sectioned) in each umbilical cord. The correlation of the number of buckles and the patency of the vessel should also therefore be limited to the small area we analyzed. i.e. an open artery with fewer folds in the area analyzed for a specific specimen may have an area with more buckles and an occluded lumen in a different region of the cord which we have not analyzed. A future, more focused study analyzing the formation of buckles along the length of the cord using a 3D imaging technique may be necessary for completely understanding the number of buckles needed for driving vessel occlusion. However, from the analysis of the cohort of cords in this study, at least 4 buckles observed in a small segment are sufficient to enable arterial occlusion. The computational modeling predicts 3-7 buckles. Since a large number of cords would be required for statistical conclusions in large mammals, and we were only able to obtain a small number, that too with difficulty, we have not done a statistical analysis on these mammalian cords.

7) Figure 5. The mutant analyses:a) In the text, legends, and figures, the notation is inconsistent for the knockout mouse models: Adamts1-/-, Adamts1 KO, Acan-/-, Acancmd/cmd, Acan mutants, and Acan KO are all used. Please choose a consistent notation for each model.

We have updated both the text and the figures to consistently use the *Adamts1^-/-^* and *Acan^-/-^*notations throughout the manuscript.

b) Please provide information on genotyping mutant litters in the Methods/Supplemental Methods, and their ratios at each embryonic day examined, especially as the authors claim that mutants did not show intrauterine growth retardation or death (see Figure 5—figure supplement 1). If possible, please indicate the background resorption level in these strains, which would be evident by having included the genotype of the resorptions.

We have added the observed genotype information as a new figure panel (Figure 4—figure supplement 1C) and rewritten the section of the manuscript better describing the *Acan^-/-^ embryos*, limiting our conclusions to their umbilical cords. Defective cartilage and impaired skeletal development the *Acan^-/-^* embryos was previously extensively characterized. Deficiencies in these processes alter embryo dimensions and we therefore have removed any conclusions or suggestions related to overall growth, and limited the focus to the umbilical cord phenotype. *Acan^-/-^* embryos are observed at the expected Mendelian ratio at E18.5 and hence do not die in utero prior to E18.5. We have not observed a higher rate of resorptions of embryos in our crossings in agreement with observations made by others.

In the initial analysis of *Acan^-/-^* mice, from a total of 733 offspring, 180 were homozygous (24.6%) as reported in Rittenhouse et al., 1978. The background strain of these mice (C57BL/6J) is reported to have an average litter size of 6.2 pups (Verley et al., 1967) and average resorption sites from embryonic day 11 to term of 1.54 + 0.15 in 3-7 month old mothers and 2.94 + 0.28 in 11-12 month old mothers (cf. Holinka, Tseng and Finch, 1979).

c) Could the authors group presentation of aggregan and adamts1 mutants rather than intermingle the data within the text? - it was difficult to follow which result correlated with which mutant.

We have rearranged this figure and grouped the *Acan* and *Adamts1* data panels (Figure 4F for *Adamts1* and Figure 4G for *Acan*) separately. The manuscript text reflects this change.

d) Figure 5G. The PHH3 staining examples are not very good. How are they being normalized to calculate a percentage?

We enlarged this figure panel and clearly marked pHH3 positive cells with white arrowheads and indicated the vessel lumen using a white dotted line (Figure 4H). The samples were counterstained with DAPI to identify all nuclei. The percentage of pHH3 positive nuclei was determined. At least two sections were stained and quantified from each umbilical cord (a total of 4 umbilical cords for each genotype (16 sections)).

8) Figure 4—figure supplement 1.

As noted above, our primary goal was to determine salient characteristics that drive umbilical artery closure at birth using a (new) computational model via parametric studies. Yet, we needed baseline passive biomechanical properties. Although we tested N=4 umbilical arteries (and N=4 veins), we previously used best-fit material parameters for a single “representative” sample. In the revised manuscript, however, we re-performed all data analysis, now based on a rigorous mean behavior (all pressure-diameter, axial force-length data for all 4 samples were combined into a single large data set and best-fit values were determined). As can be seen from the new Fig 7—figure supplement 2, the model-predicted behavior describes very well the mean responses, noting that the grey regions show standard deviations (not standard errors of the mean) to reveal the full extent of the specimen-to-specimen differences.

a) Need SD or SEM.

Since we informed the model with mean properties, we show the SD as a grey region to enable easy visual comparison of the computed mean against this backdrop.

b) The number of samples used for mechanical testing needs to be included.

Now noted, N=4 umbilical veins and N=4 umbilical arteries were analyzed, now noted.

c) Please include circumferential and axial stress-strain curves which are directly related to the fitted material parameters used for the modeling.

These figures have been added to the supplemental figure.

d) Figure 4—figure supplement 1B, subsection “Differential SMC contraction in the bilayered umbilical arteries and vein”of the text state that the artery has a smaller lumen, citing this panel, but this panel shows the outer diameter, and not the inner one.

Thank you. This has now been corrected to show the inner diameter in Figure 5—figure supplement 1.

e) Figure 4—figure supplement 1C and subsection “Differential SMC contraction in the bilayered umbilical arteries and vein”, text: Please define "distensibility" and "extensibility" as used in the text. At what pressures or axial stretches are you comparing the distensibility and extensibility? How do the axial stretch values in Figure 4—figure supplement 1C compare to the in vivo axial stretches?

Indeed, we needed to be clearer. Unfortunately, there is a clinical definition of “distensibility” that is actually a measure of structural compliance (normalized changes in diameter divided by pulse pressure). Herein, distensibility (circumferential deformation, referring to an enlargement) and extensibility (axial deformation, lengthening) are kinematic measures. Values were computed at the vessel-specific in vivo axial stretch and near physiological pressures (UA: 20 mmHg, UV: 5mmHg).

9) Figure 5—figure supplement 1.Please support the important conclusion, subsection “Aggrecan and Adamts1 are necessary for normal umbilical cord morphogenesis” based on this figure that "Neither mutant showed intrauterine growth retardation, and intrauterine death was infrequent, suggesting adequate cord circulation.", with actual measurements, especially as the acan KO mutant in Figure 5—figure supplement 1B appears smaller at E14.5 than its wildtype counterpart - assuming the same magnifications, which should be indicated.

Please see response to point 7b above. Both embryos were imaged at the same magnification and we have now added scale bars to this image. We have also limited our conclusions to umbilical cord development and removed any references and conclusions related to growth retardation of these mutants in our study. *Acan* mutants have a very abnormal skeletal system with short limbs and craniofacial defects as previously characterized, constituting an overall embryo dysmorphology and growth retardation. We observe them at the expected Mendelian ratio at E18.5 just prior to birth and our conclusions from the E14.5 embryos are limited to the reorientation of the umbilical cord SMCs which takes place from E12.5-E14.5.

10) Figure 8.- What are the blue cells? Something other than SMCs?

The blue cells represent proteoglycan-rich non-contractile SMCs that get redirected centripetally to occlude the lumen. We have modified the cartoon legend to clarify this.

11) Table 1.a) As c1 for the diagonal fibers is 2-3 orders of magnitude below the circ and axial fibers, are the diagonal fibers really necessary in the constitutive model?

It is well known that best-fit values of material parameters in exponential relations are not unique (with high c1 balancing low c2 and vice versa), hence it is most important to be consistent in the estimation (using random initial guesses, using constrained optimization to ensure non-negative values, using biaxial data, etc.) as we were. It is also important not to ascribe much physiological meaning to individual parameters in phenomenological models, but rather to focus on their collective contributions to calculated stresses, stiffnesses, energy, etc. That said, when re-fitting the data (mean data, not single representative), we found increased c1 values for the diagonal fiber families, whereas the axial family’s value of c1 decreased. The inclusion of all 4 families is, we think, prudent to ensure good fits to account for possible variability in the experimental data and is justified based on prior good fits to diverse murine data.

b) Also, the c2 values are 1-2 orders of magnitude higher than values that the group has published previously for mouse elastic arteries. Can the authors comment on the suitability of the constitutive model for fitting such nonlinear data and differences between the mechanical behavior of umbilical artery/vein and elastic arteries in the mouse?

The magnitude of the c2 values are consistent with those of some past works, cf. Supplemental Table 3 from Bersi et al., 2016 which includes some c2 values even higher than those in this work. While these nonlinear constitutive equations were initially developed for elastic arteries, they have since been used to describe the behavior of pulmonary arteries, veins, and tissue engineered vascular constructs, suggesting that they are a reasonable first approach for understanding the behavior of a new vessel. Furthermore, the key simulations in this work relate to the buckling phenomenon observed experimentally. We sought to determine whether active stress present in an external layer could drive closure and if eventual buckling of a GAG-rich inner layer was necessary for complete occlusion, which were studied parametrically. Using this constitutive approach allowed us to determine the feasibility of such a hypothesis and to understand how changes in volume related to swelling could impact the degree of contractility necessary to cause buckling.

c) Are the values in Table 1 averages of individual values from multiple arteries, fit from a combination of data from multiple arteries, or representative values from a single artery?

The initially presented values were those for a representative sample. We have since re-parameterized for the mean behavior by re-running all estimations. The value of axial stretch was also updated to 1.28

[Editors' note: further revisions were suggested prior to acceptance, as described below.]

In their revised manuscript concerning the mechanism of umbilical arterial closure at birth in Placentalia, Nandadasa et al., have satisfactorily addressed the majority of the reviewers' concerns. However, there remain two major concerns: (1) the mouse mutant analyses, and (2) the number of specimens used per experiment; a small number of minor revisions; and requested changes to the Abstract, in accord with eLife's policies.Essential revisions:1) Genetic mutants. The reviewers had requested the following (copied from the previous letter to the authors): (b) Please provide information on genotyping mutant litters in the Materials and methods section/Supplemental methods, and their ratios at each embryonic day examined, especially as the authors claim that mutants did not show intrauterine growth retardation or death (see Figure 5—figure supplement 1). If possible, please indicate the background resorption level in these strains, which would be evident by having included the genotype of the resorptions.Mouse mutant Adamts-/-. As the authors did not provide their method for genotyping this mutant strain, as requested, more investigation was needed on the part of the reviewers to understand what exactly this strain is to consider why the authors ignored the request.Background (Oller et al., 2017): The Adamts-/- mutant used in this study was described by Oller et al., 2017 as an insertion of a lacZ-bearing cassette into intron 1 of the gene. According to this previous paper, not only does this insertion reveal where Adamts1 is expressed via staining for ß-galactosidase activity, but in its hemizygous state, the insertion also causes a reduction in both mRNA and protein. In its homozygous state, ß-galactosidase activity is still detectable, but the Adamts1 protein is not (Figure 1a of Oller et al., 2017). Oller et al did not provide details on how the animals were maintained and mated (which they really should have been asked to do, alas), but they did explain how their litters were genotyped, the specific sequence used in those PCR genotyping experiments, and the genotypic ratios of their animals at weaning (Supplemental Figure 1b of Oller et al., 2017).In the current study, the authors distinguished hemizygotes and homozygotes in the figure panels (e.g., Figure 4C versus the other panels), and wildtype and homozygotes in the others that relate to the mutant analysis, implying that they genotyped this material. However, they ignored the reviewers' procedural request, which is repeated and expanded as follows:Please provide information oni) genotyping method used, including the exact DNA sequence for PCR analysis;

This (DNA isolation method, and primer sequences, expected PCR products) is now provided in detailed methods in Appendix 1

ii) genotypic ratios according to gestational day;

This data for both Acan and Adamt1s mutants is in revised Figure 4—figure supplement 1 panel C.

iii) the genetic background on which the Adamts animals were maintained (according to Oller et al., it seems to be a B/6 background, but please confirm);

We maintained them in C57BL/6, as now written in detailed Materials and methods section.

iv) the parental genotypes used to produce the specific genotypes (perhaps hemizygous by wildtype matings produced hemizygous embryos detectable by X-gal staining, whilst homozygous embryos were obtained by crossing hemizygous animals and using PCR genotyping to distinguish the three genotypes? - please confirm/clarify);

Yes, this is correct and added to Appendix 1. Either lacZ staining and genotyping were used to identify hemizygotes obtained from hemizygous X wild-type matings, whereas PCR genotyping was used to distinguish the three genotypes arising from crosses of hemizygous animals, which were used to generate homozygous embryos.

v) how gestational age was determined. Were timed matings used? if not timed matings, then how was gestational age determined?;

Yes, timed matings were used, as specified in the revised Materials and methods section.

vi) the protocol used for X-gal staining the hemizygotes.

We cite one of our previous manuscripts t where the detailed protocol was described.

- While Oller et al., 2017 used this transgenic mouse line, they did not create it. Please clarify the specific origin of this mutant mouse strain - that information was impossible to locate on the EMMA site.

These details are provided in the expanded Appendix 1 subsection “Additional details of transgenic mice”. The details of the mice are no longer listed in EMMA, unfortunately.

- It would be most helpful for the reader if the authors would introduce the Adamts mutant in the Results section by summarizing Oller et al.'s results concerning the levels of mRNA versus protein in hemizygotes and homozygotes, as under "Background", above.

This is now included in the revised results section.

- Finally, if the authors can, would they comment on whether hemizygotes exhibited foreshortened umbilical cords, too, and did their lengths fall between those of the wildtype and homozygous mutants?

We did not measure the lengths of hemizygous cords, but they were visually undistinguishable from wild-type cords.

Mouse mutant Acan-/-.- Although the authors provided the genotypic ratios of the Acan-/- mutants as requested by the reviewers (new Figure 4—figure supplement 1 Panel 4C), they did not indicate the genotyping procedure. Please add it to Detailed Methods, to include how the DNA was obtained, and the DNA sequence used to PCR the littermates' DNA.

The details of the Acan genotyping are now provided in Appendix 1.

- The mutant allele was originally described by Krueger et al., 1999, but from where did the authors procure this mouse strain?

The colony has long been established in the laboratory of Nancy Schwartz and Miriam Domowicz at the University of Chicago, and mice from that colony were transferred to the Cleveland Clinic Lerner Research Institute. Please see subsection “Additional details of transgenic mice” for details of this allele.

- Litters were obtained at E12.5, E14.5, and E18.5; please indicate how matings were carried out to ascertain the timing of gestation, including parental genotypes that produced the Acan litters.

The homozygous mutants were produced by inter-crossing hemizygous parents. We have specified this in the expanded section in Appendix 1 to explain this as part of this sentence: PCR genotyping was used to distinguish the three genotypes possibly arising from crosses of Adamts1 or Acan hemizygous animals which were used to generate homozygous embryos.

2) The reviewers had requested the number of specimens (n) for every experiment. The following are still missing:- Figure 1D. n, the number of immunostained specimens?

n=4 cords for each antibody, added to figure legend.

- Figure 2B, line 545. "n=3 umbilical cords" - for each probe?, or for both?

(a) n=3 umbilical cords for each in situ probe

(b) n=4 umbilical cords for each antibody staining

added to figure legend

- Figure 4—figure supplement 1.S4b. n = ?

n=2 Acan KO at E E12.5 and n=3 for E14.5, added to figure legend.

S4d. n = ?

n=3 UC each genotype, added to figure legend.